# Analytical Method of the Shear Lag Effect in Thin-Walled Box Girders Based on the Shear Flow Distribution Law

**Yuan Shi** [ID]**, Shijun Zhou \*, Gang Wang and Cao Zhou**

School of Civil Engineering, Chongqing University, Chongqing 400030, China; 202116021004@cqu.edu.cn (Y.S.);
gwang0401@126.com (G.W.); 202116021136t@stu.cqu.edu.cn (C.Z.)
\* Correspondence: sjzhou8@163.com

**Abstract:** This paper presents an analytical method based on the shear flow distribution law to study the shear lag effect of thin-walled single- and double-cell box girders. The first step in this method is to determine the box girder's shear flow distribution. Subsequently, a series of novel improved longitudinal displacement functions mathematically expressed as cubic parabolas are established. The parabolic origin of these functions is located at the zero point of the shear flow corresponding to each plate; the unknown parameters used to describe the function form can be determined according to the shear flow distribution, the continuity conditions, and the axial force balance condition. Then, the variational energy method is adopted to derive the governing differential equations. The shear lag effect in thin-walled single- and double-cell box girders under several boundary conditions and load cases is studied and analytical expressions for the shear lag coefficient are derived. Finally, results obtained using the proposed method are validated via comparison with numerical results. The results show that the proposed method can provide reasonable predictions for the shear lag effect of single- and double-cell box girders, and that this method is more straightforward and practical. In addition, the shear lag coefficients at different webs are not entirely equal, which is related to the distance from the web to the zero point of the shear flow.

**Keywords:** thin-walled box girders; shear lag effect; shear flow distribution; theoretical analysis; variational energy method; numerical simulation

## 1. Introduction

Thin-walled box girders are affected by a phenomenon known as the shear lag effect, which is the non-uniform stress distribution along flanges caused by shear deformation [1,2]. This effect increases deflections and peak stresses, causing bridge cracking and collapsing [3]. This topic is well known and widely studied in the literature [4–13]. Moreover, in recent decades, many scholars have adopted variational analyses to study the shear lag effect [14,15]. Although the variational energy method is simple and well applicable, its accuracy depends on the selected shear lag warping displacement functions. Many authors have contributed to this.

Reissner [14] was the first to adopt the variational energy method to study the shear lag effect and established a quadratic parabolic model to describe the normal stress distributions of rectangular box beams. However, other scholars discovered that the shear lag warping displacement function with a cubic parabola was more suitable for analyzing the shear lag effect of box girders [16–18]. Combining the above two models, Hu [19] and Yu [20] proposed that the warping displacement function of top plates was a quadratic parabola, while the warping displacement function of cantilever plates was a cubic parabola. In addition, cosine function models were also used to address the shear lag effect problem [21]. An additional term was introduced to the cosine function model to ensure the axial equilibrium of the single-cell box section [22], which improved the accuracy and preciseness of shear lag analysis. Zhang [23] then defined two global modification factors

that prevented the generation of additional axial force and bending moment caused by shear lag warping stresses, successfully capturing the self-balancing characteristic of the system. Additionally, the web's longitudinal displacement was added to satisfy axial equilibrium conditions, and independent shear lag warping displacement functions were employed in different flanges [24,25]. Zhu [26] also considered axial force balance when formulating the shear lag displacement model of the single-cell box girders and pointed out that considering axial force balance yields more precise results.

However, although there have been many assumptions regarding displacement function, little attention has been paid to analyzing the shear lag effect in box girders from the perspective of shear flow distribution, particularly in double-cell box girders. Lin [27] focused on single-cell box girders' shear flow distribution law and defined the shear lag warpage function through the shear deformation law. Li [28,29] redivided the flange width of multi-cell box girders based on the shear flow distribution before depicting the stress distribution with the cosine function. Nevertheless, the above research reviews could not obtain precisely accurate normal stress distributions, as they failed to reflect stress differences at different webs for the double-cell box section, and the process of calculating the coefficients was complex. Additionally, there is limited research on the shear lag effect in double-cell box girders. The difference in shear lag coefficients for different web plates is not reflected in [30,31]. The authors of [32] present only finite element simulation results and do not offer specific displacement functions.

This paper presents an analytical method based on the shear flow distribution law to study the shear lag effect of thin-walled single- and double-cell box girders. The first step in this method is to determine the box girder's shear flow distribution. Subsequently, a series of novel improved longitudinal displacement functions mathematically expressed as cubic parabolas are established. The parabolic origin of these functions is located at the zero point of the shear flow corresponding to each plate, and the unknown parameters used to describe the function form can be determined according to the shear flow distribution, the continuity conditions, and the axial force balance condition. Then, the variational energy method is adopted to derive the governing differential equations. The shear lag effect in thin-walled single- and double-cell box girders under several boundary conditions and load cases is studied, and analytical expressions for the shear lag coefficient are derived. Finally, the results obtained using the proposed method are validated via comparison with numerical results supplied by the ANSYS 2022 R2, a commercial finite element software.

## 2. The Distribution Law of Bending Shear Flow in Box Sections

Under vertical symmetrical loading, the bending shear flow significantly influences the normal stress distribution [33]. Therefore, the distribution pattern of the shear flow in single- and double-cell box girders was investigated in this work. Figure 1 displays simplified thin-walled box cross sections that consist of one top plate of width $2b_1$, two cantilever plates of width $b_2$, one bottom plate of width $2b_3$, and several webs of height $h$.

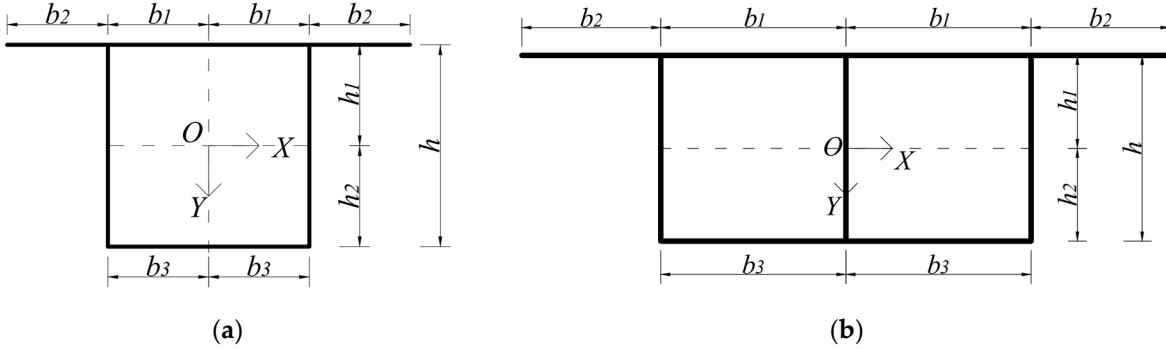

**Figure 1.** Schematic of thin-walled box sections: (**a**) single cell; (**b**) double cell.

### 2.1. Basic Coordinate System and Solution Steps

As shown in Figure 1, a coordinate system is introduced, with the origin $O$ located at the section center. The $Z$-axis aligns with the beam's longitudinal direction, the $X$-axis is along the section's width direction, and the $Y$-axis is perpendicular to the $X$- and $Z$-axis. In addition, the cross section is symmetric about the coordinate plane $Y$-$Z$.

The steps for solving the shear flow distribution of single- and double-cell box sections are organized as follows: first, the shear flow $q_0$ in the open box section is calculated; then, the additional static shear flow $q_i$ is derived using the deformation coordination relationships; finally, the above two parts are added to obtain the total shear flow distribution in the closed box section.

Notably, in-plane shear stresses are uniformly distributed along the direction of wall thickness, and the magnitude of $q_0$ is mainly determined by the static moment from the free surface to the desired point [27]. The $q_0$ can be expressed as follows:

$$q_0 = -Q(z)S_x/I_x \tag{1}$$

where $Q(z)$ represents the cross-sectional shear force; $S_x$ represents the static moment to the $X$-axis; and $I_x$ represents the inertial moment to the $X$-axis. Consider $Q(z)/I_x$ as constant 1 to ease the calculation, and then, the value of $S_x$ represents the magnitude of shear stress.

### 2.2. Shear Flow in the Single-Cell Box Girder

As can be seen from Figure 2, an opening is located at the center of the bottom plate of the thin-walled single-cell box section. According to Equation (1), $q_0$ can be obtained. Meanwhile, it is easy to find that the additional static shear flow $q_i$ is zero due to the symmetry of the single-cell box section. Consequently, the total shear flow $q$ in the closed box section equals the shear flow $q_0$ in the open box section.

Figure 2 shows the distribution pattern of shear flow in the single-cell box girder, and Table 1 lists shear flow magnitudes at critical points. The shear flow magnitude of flanges in the single-cell box section is observed to reach its maximum at intersections between webs and flanges and gradually diminishes towards the flanges' interior. Eventually, it becomes zero at the free ends of the cantilever plate and the midpoints of the top and bottom plates.

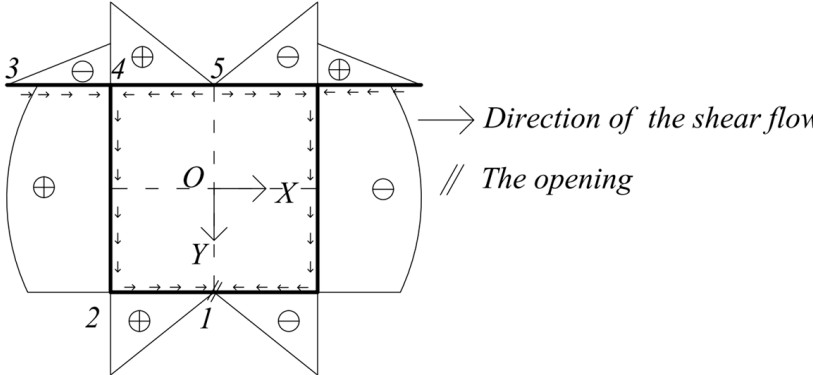

**Figure 2.** Shear flow distribution of the single-cell section.

**Table 1.** Shear flow magnitudes of the single-cell section at critical points.

| points | 1 | 2 | 3 | 4-left | 4-right | 5 |
|---|---|---|---|---|---|---|
| $q$ | 0 | $t_2 b_3 h_2$ | 0 | $-t_1 b_2 h_1$ | $t_1 b_1 h_1$ | 0 |

Notes: $t_1$ represents the thickness of the top plate and cantilever plates; $t_2$ represents the thickness of the bottom plate; $h_1$ and $h_2$ represent the distances from the center axis of the section to the mid-surface of the up and down wings, respectively.

### 2.3. Shear Flow in the Double-Cell Box Girder

Two openings are inserted at the intersection of the mid web and bottom plate of the double-cell section, and the distribution of the shear flow distribution of the open double-cell section is given, as shown in Figure 3. It should be noted that this distribution exhibits a positive and negative alternating phenomenon in the top plate. The position of the zero point can be determined as follows:

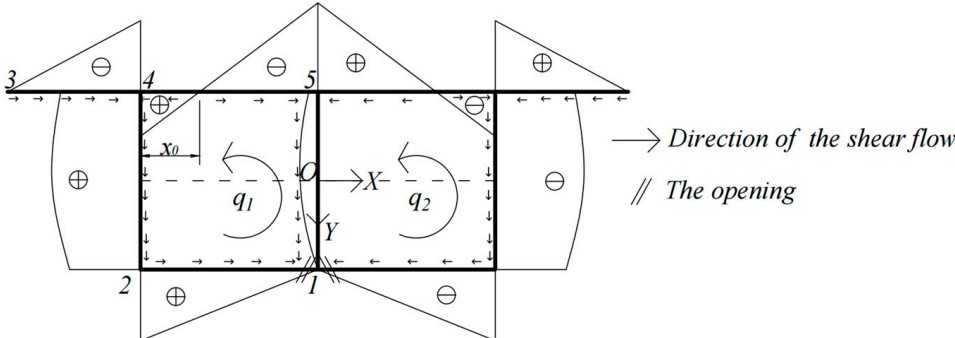

**Figure 3.** Shear flow distribution of the open double-cell section.

$$x_0 = \frac{t_2 b_3 h_2 + t_{w2} h_2^2/2 - t_{w2} h_1^2/2 - t_1 b_2 h_1}{t_1 h_1} \tag{2}$$

where $x_0$ is the distance from the zero point of the shear flow to the side web in the opened double-cell section; $tw_1$ and $tw_2$ represent the thickness of the middle web and the side web, respectively. After assigning the additional static shear flow $q_1$ and $q_2$ for the two cells, the conditions of deformation compatibility are described as follows:

$$\int_{s_1} \frac{q_0}{Gt_i} ds + \int_{s_1} \frac{q_1}{Gt_i} ds + \int_{s_1} \frac{q_2}{Gt_i} ds = 0 \tag{3}$$

$$\int_{s_2} \frac{q_0}{Gt_i} ds + \int_{s_2} \frac{q_2}{Gt_i} ds + \int_{s_2} \frac{q_1}{Gt_i} ds = 0 \tag{4}$$

where $G$ represents the shear modulus, $t$ represents the thickness, and $s$ denotes the curvilinear coordinate of the section profile. The additional shear flows can be derived as follows:

$$q_1 = -q_2 = -\frac{\dfrac{b_3^2 h_2}{2} + \dfrac{b_3 t_2 h_2 h}{t_{w2}} + \dfrac{x_0^2 h_1}{2} - \dfrac{(b_1 - x_0)^2 h_1}{2}}{\dfrac{b_3}{t_2} + \dfrac{2h}{t_{w1}} + \dfrac{b_1}{t_1} + \dfrac{h}{t_{w2}}} \tag{5}$$

Eventually, the total shear flow distribution in the closed double-cell box section can be obtained, as shown in Figure 4. The top and bottom plates exhibit a positive and negative alternating phenomenon in this distribution. It can be found that the zero points of the shear flow are not situated at the center axis of each cell but at a specific location. According to the zero points of the shear flow, the top plate is divided into two parts, the lengths of which are $b_{11}$ and $b_{12}$, and the bottom plate is divided into two parts with $b_{31}$ and $b_{32}$ lengths, respectively. Additionally, the distance from the zero point of the shear flow located at the free end of the cantilever plate to the side web is $b_{21}$. The above distances can be obtained as follows:

$$b_{11} = b_1 - b_{12} \quad b_{12} = x_0 + \frac{q_1}{t_1 h_1} \quad b_{21} = b_2 \quad b_{31} = \frac{q_1}{t_2 h_2} \quad b_{32} = b_3 - b_{31} \tag{6}$$

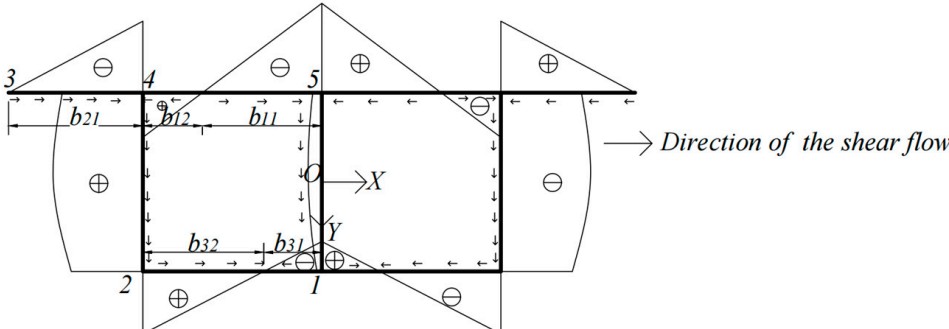

**Figure 4.** Shear flow distribution of the closed double-cell section.

Table 2 lists the shear flow magnitudes at critical points in the closed double-cell section.

**Table 2.** Shear flow magnitudes of the closed double-cell section at critical points.

| points | 1-left | 2 | 3 | 4-left | 4-right | 5 |
|--------|--------|---|---|--------|---------|---|
| $q$ | $-t_2b_{31}h_2$ | $t_2b_{32}h_2$ | 0 | $-t_1b_{21}h_1$ | $t_1b_{12}h_1$ | $-t_1b_{11}h_1$ |

## 3. Box Girder's Longitudinal Displacement Functions

Based on the shear flow distribution law obtained in the previous section, this section established the expressions of longitudinal warping displacement functions of single- and double-cell box girders. Three different analytical perspectives were proposed to address the coefficients introduced before the cubic term, and constant terms were deduced.

### 3.1. Basic Assumptions

Some assumptions are reasonably introduced before constructing displacement function models: (1) the neutral axis of thin-walled box beam sections remains constant under bending stresses; (2) the box girders are in the linear elastic stage, and the shear deformation out of the plane is neglected; and (3) the shear lag effect only alters the normal stress distribution in the cross section, with no effect on the distribution of internal forces along the beam's longitudinal direction [34].

Based on the above considerations, the box girder's longitudinal displacement function is expressed as follows:

$$u(x,z) = -yw'(z) - yf(x)\varphi(z) \tag{7}$$

where $u(x,z)$ is the longitudinal displacement; $w(z)$ is the vertical deflection; $f(x)$ is a distribution function corresponding to the shear lag effect; and $\varphi(z)$ is the maximum difference in the shear angle.

In addition, the shear lag effect of the box girder can be regarded as a plane stress problem [35], and the relationship between displacement and strain can be written as follows:

$$\varepsilon = \frac{\partial u}{\partial z} \qquad \gamma = \frac{\partial u}{\partial x} \tag{8}$$

where $\varepsilon$ is the axial strain and $\gamma$ is the shear strain. The bending normal stress is described as follows:

$$\sigma = E\frac{\partial u}{\partial z} \tag{9}$$

where $\sigma$ is the bending normal stress and $E$ is the elastic modulus.

### 3.2. Displacement Functions of the Single-Cell Box Girder

Considering the warping of webs, longitudinal displacement functions with their origins being on the zero points of the shear flow in the single-cell box girder are established as follows:

$$
\begin{cases}
u_{11} = h_1 \left[ w'(z) + \left( \eta_1 \dfrac{x^3}{b_1^3} + d_1 \right) \varphi(z) \right] & (0 \le x \le b_1) \\[3mm]
u_{12} = h_1 \left[ w'(z) + \left( -\eta_2 \dfrac{(x - b_1 - b_2)^3}{b_2^3} + d_2 \right) \varphi(z) \right] & (b_1 \le x \le b_1 + b_2) \\[3mm]
u_{13} = -h_2 \left[ w'(z) + \left( \eta_3 \dfrac{x^3}{b_3^3} + d_3 \right) \varphi(z) \right] & (0 \le x \le b_3) \\[3mm]
u_{1w2} = -y w'(z) + h_1 (\eta_1 + d_1) \varphi(z) & (-h_1 \le y \le h_2)
\end{cases}
\tag{10}
$$

where $u_{1i}$ represents the longitudinal displacement function of single-cell box girder's slabs: $i = 1$ for the top plate, $i = 2$ for the cantilever plate, $i = 3$ for the bottom plate, and $i = w_2$ for the side web; $\eta_i$ and $d_i$ are introduced coefficients. The term $\eta_i$ can be selected from three different analytical perspectives:

(1) The first approach for selecting the <u>coefficients</u> for <u>single-cell</u> box girders (1CS) is based on the classical cubic parabola formula [17]. The $\eta_i$ can be expressed as follows:

$$
\eta_1 = \eta_2 = \eta_3 = 1
\tag{11}
$$

(2) The second approach of selecting coefficients (2CS) involves using the relative magnitudes of the shear flow, which can be derived from Table 1. This is mainly due to differences in the maximum shear flow in different flanges. The $\eta_i$ can be expressed as follows:

$$
\eta_1 = \frac{b_1 t_1 h_1}{b_1 t_1 h_1} \qquad \eta_2 = \frac{b_2 t_1 h_1}{b_1 t_1 h_1} \qquad \eta_3 = \frac{b_3 t_2 h_2}{b_1 t_1 h_1}
\tag{12}
$$

(3) The third approach of selecting coefficients (3CS) involves using the relative shear deformation of each flange. The $\eta_i$ can be expressed as follows:

$$
\eta_1 = \frac{b_1^2 t_1 h_1}{b_1^2 t_1 h_1} \qquad \eta_2 = \frac{b_2^2 t_1 h_1}{b_1^2 t_1 h_1} \qquad \eta_3 = \frac{b_3^2 t_2 h_2}{b_1^2 t_1 h_1}
\tag{13}
$$

In addition, because the axial force caused by the shear lag effect is zero [24] and the displacements and stresses are continuous. The expressions for $d_i$ are derived as follows:

$$
\begin{aligned}
\oint \sigma_1 dA &= 2E\varphi'(z) \cdot
\begin{bmatrix}
\int_0^{b_1} t_1 h_1 \left( \eta_1 \dfrac{x^3}{b_1^3} + d_1 \right) dx + \int_{b_1}^{b_1+b_2} t_1 h_1 \left( -\eta_2 \dfrac{(x - b_1 - b_2)^3}{b_2^3} + d_2 \right) dx \\[3mm]
-\int_0^{b_3} t_2 h_2 \left( \eta_3 \dfrac{x^3}{b_3^3} + d_3 \right) dx + \int_{-h_1}^{h_2} t_{w2} h_1 (\eta_1 + d_1) dx
\end{bmatrix} \\[3mm]
&= 2E\varphi'(z) \cdot
\begin{bmatrix}
t_1 h_1 b_1 \left( \dfrac{\eta_1}{4} + d_1 \right) + t_1 h_1 b_2 \left( \dfrac{\eta_2}{4} + d_2 \right) \\[3mm]
-t_2 h_2 b_3 \left( \dfrac{\eta_3}{4} + d_3 \right) + t_{w2} (\eta_1 + d_1) h h_1
\end{bmatrix} = 0
\end{aligned}
\tag{14}
$$

$$
\begin{cases}
d_1 = -(A_1 h_1 \eta_1 + 4A_2 h_1 \eta_1 - 3A_2 h_1 \eta_2 + 4A_3 h_1 \eta_1 + 3A_3 h_2 \eta_3 + 4A_w h_1 \eta_1)/(4h_1 A) \\[3mm]
d_2 = \eta_1 + d_1 - \eta_2 \qquad\qquad d_3 = -\dfrac{h_1}{h_2}(\eta_1 + d_1) - \eta_3
\end{cases}
\tag{15}
$$

where $A_1$ denotes the area of the top plate, $A_2$ denotes the area of the cantilever plate; $A_3$ denotes the area of the bottom plate; $A_w$ denotes the area of the web plate; and $A$ denotes the total cross-sectional area.

### 3.3. Displacement Functions of the Double-Cell Box Girder

Similarly to single-cell box girders, the longitudinal displacement functions of double-cell box girders are constructed as follows:

$$
\begin{cases}
u_{21} = h_1 \left[ w'(z) + \left( -\eta_1 \dfrac{(x - b_{11})^3}{b_{11}^3} + d_1 \right) \varphi(z) \right] & (0 \leq x \leq b_{11}) \\[3mm]
u_{22} = h_1 \left[ w'(z) + \left( \eta_2 \dfrac{(x - b_{11})^3}{b_{12}^3} + d_2 \right) \varphi(z) \right] & (b_{11} \leq x \leq b_1) \\[3mm]
u_{23} = h_1 \left[ w'(z) + \left( -\eta_3 \dfrac{(x - b_1 - b_{21})^3}{b_{21}^3} + d_3 \right) \varphi(z) \right] & (b_1 \leq x \leq b_1 + b_2) \\[3mm]
u_{24} = -h_2 \left[ w'(z) + \left( -\eta_4 \dfrac{(x - b_{31})^3}{b_{31}^3} + d_4 \right) \varphi(z) \right] & (0 \leq x \leq b_{31}) \\[3mm]
u_{25} = -h_2 \left[ w'(z) + \left( \eta_5 \dfrac{(x - b_{31})^3}{b_{32}^3} + d_5 \right) \varphi(z) \right] & (b_{31} \leq x \leq b_3) \\[3mm]
u_{2w1} = -y\omega'(z) + h_1(\eta_1 + d_1)\varphi(z) & (-h_1 \leq y \leq h_2) \\[3mm]
u_{2w2} = -y\omega'(z) + h_1(\eta_2 + d_2)\varphi(z) & (-h_1 \leq y \leq h_2)
\end{cases} \tag{16}
$$

where $u_{2i}$ represents the longitudinal displacement function of double-cell box girder's slabs: $i = 1$ for the top plate of length $b_{11}$, $i = 2$ for the top plate of length $b_{12}$, $i = 3$ for the cantilever plate of length $b_{21}$, $i = 4$ for the bottom plate of length $b_{31}$, $i = 5$ for the bottom plate of length $b_{32}$, $i = w_1$ for the middle web, and $i = w_2$ for the side web; $\eta_i$ and $d_i$ are introduced coefficients. The term $\eta_i$ can also be set from the same perspectives as the single-cell box girder:

(1) In the <u>first</u> approach for selecting the <u>coefficients</u> for <u>double-cell</u> box girders (1*CD*), the $\eta_i$ can be expressed as follows:

$$
\eta_1 = \eta_2 = \eta_3 = \eta_4 = \eta_5 = 1 \tag{17}
$$

(2) In the second approach (2*CD*), using Table 2, the $\eta_i$ can be expressed as follows:

$$
\eta_1 = \frac{b_{11}t_1h_1}{b_{11}t_1h_1} \quad \eta_2 = \frac{b_{12}t_1h_1}{b_{11}t_1h_1} \quad \eta_3 = \frac{b_{21}t_1h_1}{b_{11}t_1h_1} \quad \eta_4 = \frac{b_{31}t_2h_2}{b_{11}t_1h_1} \quad \eta_5 = \frac{b_{32}t_2h_2}{b_{11}t_1h_1} \tag{18}
$$

(3) In the <u>third</u> approach (3*CD*), the $\eta_i$ can be expressed as follows:

$$
\eta_1 = \frac{b_{11}^2 t_1 h_1}{b_{11}^2 t_1 h_1} \quad \eta_2 = \frac{b_{12}^2 t_1 h_1}{b_{11}^2 t_1 h_1} \quad \eta_3 = \frac{b_{21}^2 t_1 h_1}{b_{11}^2 t_1 h_1} \quad \eta_4 = \frac{b_{31}^2 t_2 h_2}{b_{11}^2 t_1 h_1} \quad \eta_5 = \frac{b_{32}^2 t_2 h_2}{b_{11}^2 t_1 h_1} \tag{19}
$$

However, when considering the continuity condition, a contradiction emerges:

$$
d_5 = -\frac{h_1}{h_2}(\eta_1 + d_1) - \eta_4 \qquad d_5 = -\frac{h_1}{h_2}(\eta_2 + d_1) - \eta_5 \tag{20}
$$

Therefore, it is necessary to correct $\eta_5$, which is rewritten as follows:

$$
\eta_5 = \frac{h_1}{h_2}(\eta_1 - \eta_2) + \eta_4 \tag{21}
$$

The coefficients $d_i$ are derived as follows:

$$\frac{\oint \sigma_1 dA}{2E\varphi'(z)} = \begin{bmatrix} \int_0^{b_{11}} t_1 h_1 \left( -\eta_1 \frac{(x-b_{11})^3}{b_{11}^3} + d_1 \right) dx + \int_{b_{11}}^{b_{11}+b_{12}} t_1 h_1 \left( \eta_2 \frac{(x-b_{11})^3}{b_{12}^3} + d_2 \right) dx \\ + \int_{b_{11}+b_{12}}^{b_{11}+b_{12}+b_{21}} t_1 h_1 \left( -\eta_3 \frac{(x-b_{11}-b_{12}-b_{21})^3}{b_{21}^3} + d_3 \right) dx \\ - \int_0^{b_{31}} t_2 h_2 \left( -\eta_4 \frac{(x-b_{31})^3}{b_{31}^3} + d_4 \right) dx - \int_{b_{31}}^{b_{31}+b_{32}} t_2 h_2 \left( \eta_5 \frac{(x-b_{31})^3}{b_{32}^3} + d_5 \right) dx \\ + \frac{1}{2} \int_{-h_1}^{h_2} t_{w1} h_1 (\eta_1 + d_1) dx + \int_{-h_1}^{h_2} t_{w2} h_1 (\eta_2 + d_2) dx \end{bmatrix}$$

$$= \begin{bmatrix} t_1 h_1 \left( b_{11} \left( \frac{\eta_1}{4} + d_1 \right) + b_{12} \left( \frac{\eta_2}{4} + d_2 \right) + b_{21} \left( \frac{\eta_3}{4} + d_3 \right) \right) + \frac{1}{2} t_{w1} h_1 h (\eta_1 + d_1) \\ -t_2 h_2 \left( b_{31} \left( \frac{\eta_4}{4} + d_4 \right) + b_{32} \left( \frac{\eta_5}{4} + d_5 \right) \right) + t_{w2} h_1 h (\eta_2 + d_2) \end{bmatrix} = 0 \tag{22}$$

$$\begin{cases} d_1 = -\dfrac{A_{11}\eta_1 + A_{12}\eta_2 - 3\eta_3 A_2 + (3\eta_1 + \eta_2)A_{32} + \dfrac{3\eta_4 h_2 A_3}{h_1}}{4A} - \dfrac{\eta_2 A_2 + \eta_1 A_{31} + \eta_1 A_{w1} + \eta_2 A_{w2}}{A} \\ d_2 = d_1 \qquad d_3 = \eta_2 + d_2 - \eta_3 \qquad d_4 = -\dfrac{h_1}{h_2}(\eta_1 + d_1) - \eta_4 \qquad d_4 = d_5 \end{cases} \tag{23}$$

where $A_{ij}$ denotes the corresponding plate's area.

## 4. Governing Differential Equations and Boundary Conditions

The governing differential equations and boundary conditions for box girders can be derived using the principle of minimum potential energy. This principle is expressed as follows [7]:

$$\delta\Pi = \delta(V + U) = 0 \tag{24}$$

where $\Pi$ denotes the total potential energy; $V$ denotes the external load potential energy; and $U$ denotes the strain energy. Under the vertical load, the $V$ and $U$ of each slab can be expressed as follows:

$$V = \int_0^L M(z)w''(z)dz \tag{25}$$

$$U = \frac{1}{2} \iint t \left( E\varepsilon^2 + G\gamma^2 \right) dy dz \tag{26}$$

where $L$ represents the length of the beam; $M(z)$ represents the bending moment of the cross section. Substituting Equation (8) into Equation (26), the strain energy can be rewritten as follows:

$$U = \frac{1}{2} \iint t \left( E \left( \frac{\partial u}{\partial z} \right)^2 + G \left( \frac{\partial u}{\partial x} \right)^2 \right) dy dz \tag{27}$$

For the single-cell box girder, substituting Equation (10) into Equation (27), the strain energy of side webs $U_{1w2}$ is expressed as follows:

$$U_{1w2} = \frac{1}{2} EI_{1w2} \int_0^L \left[ (w'')^2 - \frac{3h_1(h_2 - h_1)(\eta_1 + d_1)}{h_2^2 - h_1 h_2 + h_1^2} w'' \varphi' + \frac{3h_1^2(\eta_1 + d_1)^2}{h_2^2 - h_1 h_2 + h_1^2} (\varphi')^2 \right] dz \tag{28}$$

where $I_{1w2} = 2t_{1w2} \left( h_2^3 + h_1^3 \right) / 3$ is the moment of inertia of side webs. The strain energy of each flange $U_{1i}$ is expressed as follows:

$$U_{1i} = \frac{1}{2} EI_{1i} \int_0^L \left[ (w'')^2 + 2w'' \varphi' \left( \frac{\eta_i}{4} + d_i \right) + (\varphi')^2 \left( \frac{\eta_i^2}{7} + \frac{\eta_i d_i}{2} + d_i^2 \right) + \frac{9G\eta_i^2}{5Eb_k^2} \varphi^2 \right] dz \tag{29}$$

where $I_{1i} = 2b_k t_k h_k^2$ is the inertia moment of each flange after ignoring its self-inertia moment; $t_k$ is the thickness; $h_k$ is the distance from each plate to the neutral axis; and $b_k$ is the width.

For the double-cell box girder, substituting Equation (16) into Equation (27), the strain energy of the middle web $U_{2w1}$ is expressed as follows:

$$U_{2w1} = \frac{1}{2} E I_{2w1} \int_0^L \left[ (w'')^2 - \frac{3h_1(h_2 - h_1)(\eta_1 + d_1)}{h_2^2 - h_1 h_2 + h_1^2} w'' \varphi' + \frac{3h_1^2(\eta_1 + d_1)^2}{h_2^2 - h_1 h_2 + h_1^2} (\varphi')^2 \right] dz \tag{30}$$

where $I_{2w1} = t_{w1}\left(h_2^3 + h_1^3\right)/3$ is the moment of inertia of the middle web. The strain energy of side webs $U_{2w2}$ is expressed as follows:

$$U_{2w2} = \frac{1}{2} E I_{2w2} \int_0^L \left[ (w'')^2 - \frac{3h_1(h_2 - h_1)(\eta_2 + d_2)}{h_2^2 - h_1 h_2 + h_1^2} w'' \varphi' + \frac{3h_1^2(\eta_2 + d_2)^2}{h_2^2 - h_1 h_2 + h_1^2} (\varphi')^2 \right] dz \tag{31}$$

where $I_{2w2} = 2t_{1w2}\left(h_2^3 + h_1^3\right)/3$ is the moment of inertia of side webs. The strain energy of each flange $U_{2i}$ is expressed as follows:

$$U_{2i} = \frac{1}{2} E I_{2i} \int_0^L \left[ (w'')^2 + 2w'' \varphi' \left( \frac{\eta_i}{4} + d_i \right) + (\varphi')^2 \left( \frac{\eta_i^2}{7} + \frac{\eta_i d_i}{2} + d_i^2 \right) + \frac{9G\eta_i^2}{5Eb_k^2} \varphi^2 \right] dz \tag{32}$$

where $I_{2i} = 2b_k t_k h_k^2$ is the inertia moment of each flange after ignoring its self-inertia moment.

The total strain energy $U$ of the system can be obtained by adding the strain energy of each plate. Subsequently, the system's total potential energy $\Pi$ is expressed as follows:

$$\Pi = U + V = \int_0^L \left[ Mw'' + \frac{E}{2} N_1 (w'')^2 + E N_2 w'' \varphi' + \frac{E}{2} N_3 (\varphi')^2 + \frac{9G}{10} N_4 \varphi^2 \right] dz \tag{33}$$

where $N_i$ is the parameter related to the cross-sectional properties.

For single-cell box girders, the parameters $N_i$ can be expressed as follows:

$$\begin{cases} N_1 = \sum_{i=1}^{3} I_{1i} + I_{1w2} & N_2 = \sum_{i=1}^{3} I_{1i}\left( \frac{\eta_i}{4} + d_i \right) - \frac{3h_1(h_2 - h_1)(\eta_1 + d_1)}{2\left(h_2^2 - h_1 h_2 + h_1^2\right)} I_{1w2} \\ N_3 = \sum_{i=1}^{3} I_{1i}\left( \frac{\eta_i^2}{7} + \frac{\eta_i d_i}{2} + d_i^2 \right) + \frac{3h_1^2(\eta_1 + d_1)^2}{h_2^2 - h_1 h_2 + h_1^2} I_{1w2} & N_4 = \sum_{i=1}^{3} I_{1i} \frac{\eta_i^2}{b_k^2} \end{cases} \tag{34}$$

For double-cell box girders, the parameters $N_i$ can be expressed as follows:

$$\begin{cases} N_1 = \sum_{i=1}^{5} I_{2i} + \sum_{i=1}^{2} I_{2wi} & N_2 = \sum_{i=1}^{5} I_{2i}\left( \frac{\eta_i}{4} + d_i \right) - \sum_{i=1}^{2} \frac{3h_1(h_2 - h_1)}{2\left(h_2^2 - h_1 h_2 + h_1^2\right)}(\eta_i + d_i) I_{2wi} \\ N_3 = \sum_{i=1}^{5} I_{2i}\left( \frac{\eta_i^2}{7} + \frac{\eta_i d_i}{2} + d_i^2 \right) + \frac{3h_1^2(\eta_1 + d_1)^2}{\left(h_2^2 - h_1 h_2 + h_1^2\right)} I_{2w1} + \frac{3h_1^2(\eta_2 + d_2)^2}{\left(h_2^2 - h_1 h_2 + h_1^2\right)} I_{2w2} \\ N_4 = \sum_{i=1}^{5} I_{2i} \frac{\eta_i^2}{b_k^2} \end{cases} \tag{35}$$

Substituting Equation (33) into Equation (24), and then applying the partial integration method, governing differential equations and boundary conditions are deduced as follows:

$$\left[ E N_1 w'' + E N_2 \varphi' + M \right] \delta w'' = 0 \tag{36}$$

$$\left[ E N_2 w''' + E N_3 \varphi'' - \frac{9G}{5} N_4 \varphi \right] \delta \varphi = 0 \tag{37}$$

$$\left[ E N_2 w'' + E N_3 \varphi' \right] \delta \varphi \big|_0^L = 0 \tag{38}$$

According to Equations (36) and (37), the differential equation of the maximum difference in the shear angle $\varphi$ can be expressed as follows:

$$\varphi'' - \alpha^2 \varphi = \beta Q(z) \tag{39}$$

In which,

$$\alpha = \sqrt{\frac{9GN_1N_4}{5E\left(N_1N_3 - N_2^2\right)}} \qquad \beta = \frac{N_2}{E\left(N_1N_3 - N_2^2\right)} \qquad Q(z) = M'(z) \tag{40}$$

## 5. Shear Lag Coefficient

As an essential indicator of the shear lag effect, the shear lag coefficient $\lambda$ represents the relative relationship between the stress considering the shear lag effect and the stress derived by the elementary beam theory. The shear lag coefficient can be defined as follows:

$$\lambda = \sigma/\sigma_0 = 1 + \sigma_1/\sigma_0 \tag{41}$$

where $\sigma_0$ is the stress calculated by the primary beam theory and $\sigma_1$ is the stress caused by the shear lag. In addition, the phenomenon that $\lambda$ is has a value greater than one at the web and less than one at the flanges' interior is called the positive shear lag effect, the opposite of which is the negative shear lag effect [36].

As shown in Figure 5, $a_{11}$ and $a_{22}$ represent the zero points of the shear flow of the top slab; $a_{12}$, $a_{21}$, and $a_{23}$ stand for the junctions of the top slab and webs; $a_{13}$ and $a_{24}$ represent the zero points of the shear flow of the cantilever slab; $a_{14}$ and $a_{26}$ represent the zero points of the shear flow of the bottom slab; and $a_{15}$, $a_{25}$, and $a_{27}$ stand the for the junctions of the bottom slab and webs. These points are the maximum or minimum stress points on each plate, so their shear lag coefficients are representative.

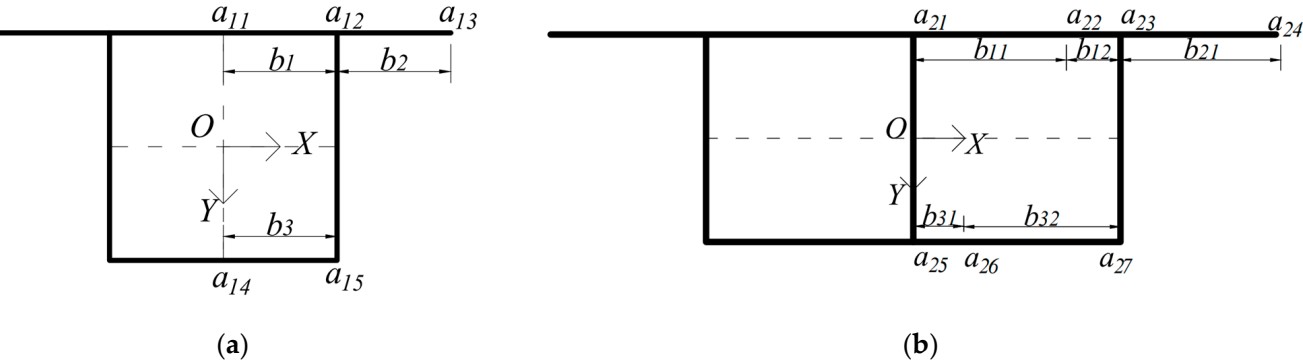

**(a)**                        **(b)**

**Figure 5.** Schematic of critical points on the box section: (**a**) single cell; (**b**) double cell.

For single-cell box girders, the shear lag coefficients of critical points can be expressed as follows:

$$\begin{cases} \lambda(a_{11}) = 1 - \dfrac{(N_1d_1 - N_2)E}{M}\varphi' & \lambda(a_{12}) = 1 - \dfrac{(N_1\eta_1 + N_1d_1 - N_2)E}{M}\varphi' \\[2mm] \lambda(a_{13}) = 1 - \dfrac{(N_1d_2 - N_2)E}{M}\varphi' & \lambda(a_{14}) = 1 - \dfrac{(N_1d_3 - N_2)E}{M}\varphi' \\[2mm] \lambda(a_{15}) = 1 - \dfrac{(N_1\eta_3 + N_1d_3 - N_2)E}{M}\varphi' \end{cases} \tag{42}$$

For double-cell box girders, the shear lag coefficients of critical points can be expressed as follows:

$$\begin{cases} \lambda(a_{21}) = 1 - \dfrac{(N_1\eta_1 + N_1d_1 - N_2)E}{M}\varphi' \quad & \lambda(a_{22}) = 1 - \dfrac{(N_1d_1 - N_2)E}{M}\varphi' \\[2mm] \lambda(a_{23}) = 1 - \dfrac{(N_1\eta_2 + N_1d_2 - N_2)E}{M}\varphi' \quad & \lambda(a_{24}) = 1 - \dfrac{(N_1d_3 - N_2)E}{M}\varphi' \\[2mm] \lambda(a_{25}) = 1 - \dfrac{(N_1\eta_4 + N_1d_4 - N_2)E}{M}\varphi' \quad & \lambda(a_{26}) = 1 - \dfrac{(N_1d_4 - N_2)E}{M}\varphi' \\[2mm] \lambda(a_{27}) = 1 - \dfrac{(N_1\eta_5 + N_1d_5 - N_2)E}{M}\varphi' \end{cases} \tag{43}$$

## 6. Closed Solutions of the Shear Lag Effect under Several Common Boundaries and Loads

Generally, shear force $Q(z)$ is linearly distributed in bridge structures. Analytical solutions for the maximum shear angle and the vertical deflection are derived from Equation (39) and are expressed as follows:

$$\varphi = \beta\left(C_1 \sinh \alpha z + C_2 \cosh \alpha z - \frac{Q(z)}{\alpha^2}\right) \tag{44}$$

$$\varphi' = \beta\alpha\left(C_1 \cosh \alpha z + C_2 \sinh \alpha z - \frac{Q'(z)}{\alpha^3}\right) \tag{45}$$

$$w' = -\frac{\int M(z)dz}{EN_1} - \frac{N_2}{N_1}\beta\left(C_1 \sinh \alpha z + C_2 \cosh \alpha z - \frac{Q(z)}{\alpha^2}\right) + C_3 \tag{46}$$

$$w = -\frac{\iint M(z)dz}{EN_1} - \frac{N_2\beta}{N_1\alpha}\left(C_1 \cosh \alpha z + C_2 \sinh \alpha z - \frac{M(z)}{\alpha}\right) + C_3 z + C_4 \tag{47}$$

where $C_i$ is the relevant unknown coefficient determined by the boundary and continuity conditions.

According to Equations (36) and (37), several common boundary conditions are listed: (1) When the beam is fixed, $w = 0$, $w' = 0$, and $\varphi = 0$; (2) When the beam is hinged, $w = 0$ and $\varphi' = 0$; (3) When the beam is free, $\varphi' = 0$. Additionally, analytical solutions for the shear lag effect of four typical bridge structural systems frequently used in engineering are presented.

### 6.1. Simply Supported Beam under Concentrated Load

As depicted in Figure 6, the span length of the simply supported beam is $L$, the left span length is $l_1$, and the right span length is $l_2$. In addition, a concentrated load $F$ is applied at an arbitrary position of the simply supported beam. The analytical solutions of this system are described as follows:

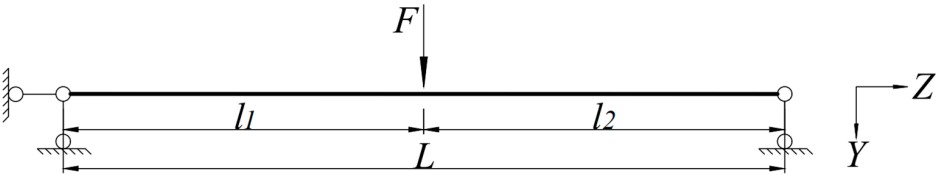

**Figure 6.** Simply supported beam under concentrated load.

$$\varphi = \begin{cases} \beta\left(C_{11} \sinh \alpha z + C_{21} \cosh \alpha z - \dfrac{bF}{L\alpha^2}\right)(0 \leq z \leq a) \\[3mm] \beta\left(C_{12} \sinh \alpha z + C_{22} \cosh \alpha z + \dfrac{aF}{L\alpha^2}\right)(a \leq z \leq L) \end{cases} \tag{48}$$

$$\varphi' = \begin{cases} \beta\alpha(C_{11} \cosh \alpha z + C_{21} \sinh \alpha z)(0 \leq z \leq a) \\ \beta\alpha(C_{12} \cosh \alpha z + C_{22} \sinh \alpha z)(a \leq z \leq L) \end{cases} \tag{49}$$

$$w' = \begin{cases} -\dfrac{bFz^2}{2LEN_1} - \dfrac{N_2\beta}{N_1}\left(C_{11}\sinh\alpha z + C_{21}\cosh\alpha z - \dfrac{bF}{\alpha^2 L}\right) + C_{31} & (0 \le z \le a) \\[3mm] \dfrac{aF(L-z)^2}{2LEN_1} - \dfrac{N_2\beta}{N_1}\left(C_{12}\sinh\alpha z + C_{22}\cosh\alpha z + \dfrac{aF}{\alpha^2 L}\right) + C_{32} & (a \le z \le L) \end{cases} \tag{50}$$

$$w = \begin{cases} -\dfrac{bFz^3}{6LEN_1} - \dfrac{N_2\beta}{N_1\alpha}\left(C_{11}\cosh\alpha z + C_{21}\sinh\alpha z - \dfrac{bFz}{\alpha L}\right) + C_{31}z + C_{41} & (0 \le z \le a) \\[3mm] -\dfrac{aF(L-z)^3}{6LEN_1} - \dfrac{N_2\beta}{N_1\alpha}\left(C_{12}\cosh\alpha z + C_{22}\sinh\alpha z - \dfrac{aF(L-z)}{\alpha L}\right) + C_{32}z + C_{42} & (a \le z \le L) \end{cases} \tag{51}$$

in which

$$\begin{cases} C_{11} = 0 \quad C_{12} = \dfrac{F\sinh\alpha a}{\alpha^2} \quad C_{21} = \dfrac{F\sinh\alpha b}{\alpha^2\sinh\alpha L} \quad C_{22} = -\dfrac{F\cosh\alpha L\sinh\alpha a}{\alpha^2\sinh\alpha L} \\[3mm] C_{31} = \dfrac{2aL^2 - 3a^2 L + a^3}{6LEN_1}F \quad C_{32} = \dfrac{a^3 - aL^2}{6LEN_1}F \quad C_{41} = 0 \quad C_{42} = \dfrac{aL^2 - a^3}{6EN_1}F \end{cases} \tag{52}$$

### 6.2. Simply Supported Beam under Uniformly Distributed Load

As depicted in Figure 7, the uniformly distributed load $f$ is applied to the simply supported beam. The analytical solutions of this system are described as follows:

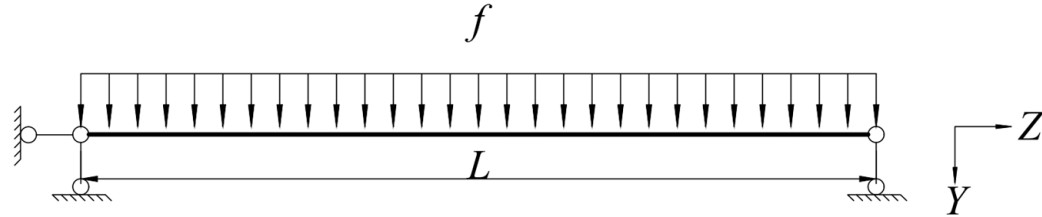

**Figure 7.** Simply supported beam under uniformly distributed load.

$$\varphi = \beta\left(C_1\sinh\alpha z + C_2\cosh\alpha z - \frac{L-2z}{2\alpha^2}f\right) \tag{53}$$

$$\varphi' = \beta\alpha\left(C_1\cosh\alpha z + C_2\sinh\alpha z + \frac{f}{\alpha^3}\right) \tag{54}$$

$$w' = -\frac{3Lz^2 - 2z^3}{12EN_1}f - \frac{N_2}{N_1}\beta\left(C_1\sinh\alpha z + C_2\cosh\alpha z - \frac{L-2z}{2\alpha^2}f\right) + C_3 \tag{55}$$

$$w = -\frac{2Lz^3 - z^4}{24EN_1}f - \frac{N_2\beta}{N_1\alpha}\left(C_1\cosh\alpha z + C_2\sinh\alpha z - \frac{Lz - z^2}{2\alpha}f\right) + C_3z + C_4 \tag{56}$$

in which

$$\begin{cases} C_1 = -\dfrac{f}{\alpha^3} \quad C_2 = \dfrac{f}{\alpha^3\sinh\alpha L}(\cosh\alpha L - 1) \\[3mm] C_3 = \dfrac{L^3 f}{24EN_1} \quad C_4 = -\dfrac{N_2\beta f}{N_1\alpha^4} \end{cases} \tag{57}$$

### 6.3. Cantilever Beam under Concentrated Load

As shown in Figure 8, the concentrated load $F$ is applied at the free end of the cantilever beam. The analytical solutions of this system are described as follows:

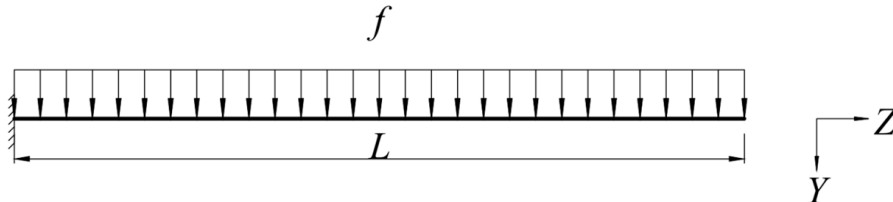

**Figure 8.** Cantilever beam under concentrated load.

$$\varphi = \beta \left( C_1 \sinh \alpha z + C_2 \cosh \alpha z - \frac{F}{\alpha^2} \right) \tag{58}$$

$$\varphi' = \beta \alpha (C_1 \cosh \alpha z + C_2 \sinh \alpha z) \tag{59}$$

$$w' = -\frac{F(z-L)^2}{2EN_1} - \frac{N_2}{N_1}\beta \left( C_1 \sinh \alpha z + C_2 \cosh \alpha z - \frac{F}{\alpha^2} \right) + C_3 \tag{60}$$

$$w = -\frac{F(z-L)^3}{6EN_1} - \frac{N_2\beta}{N_1\alpha} \left( C_1 \cosh \alpha z + C_2 \sinh \alpha z - \frac{F(z-L)}{\alpha} \right) + C_3 z + C_4 \tag{61}$$

in which

$$\begin{cases} C_1 = -\dfrac{F \sinh \alpha L}{\alpha^2 \cosh \alpha L} \qquad C_2 = \dfrac{F}{\alpha^2} \qquad C_3 = \dfrac{FL^2}{2EN_1} \\[2mm] C_4 = \dfrac{N_2\beta F(\alpha L \cosh \alpha L - \sinh \alpha L)}{N_1 \alpha^3 \cosh \alpha L} - \dfrac{FL^3}{6EN_1} \end{cases} \tag{62}$$

*6.4. Cantilever Beam under Uniformly Distributed Load*

As shown in Figure 9, the uniformly distributed load $f$ is applied to the cantilever beam. The analytical solutions of this system are described as follows:

**Figure 9.** Cantilever beam under uniformly distributed load.

$$\varphi = \beta \left( C_1 \sinh \alpha z + C_2 \cosh \alpha z - \frac{f(L-z)}{\alpha^2} \right) \tag{63}$$

$$\varphi' = \beta \alpha \left( C_1 \cosh \alpha z + C_2 \sinh \alpha z + \frac{f}{\alpha^3} \right) \tag{64}$$

$$w' = -\frac{f(L-z)^3}{6EN_1} - \frac{N_2}{N_1}\beta \left( C_1 \sinh \alpha z + C_2 \cosh \alpha z - \frac{f(L-z)}{\alpha^2} \right) + C_3 \tag{65}$$

$$w = \frac{f(L-z)^4}{24EN_1} - \frac{N_2\beta}{N_1\alpha} \left( C_1 \cosh \alpha z + C_2 \sinh \alpha z + \frac{f(L-z)^2}{2\alpha} \right) + C_3 z + C_4 \tag{66}$$

in which

$$\begin{cases} C_1 = -\dfrac{(\alpha L \sinh \alpha L + 1)f}{\alpha^3 \cosh \alpha L} \qquad C_2 = \dfrac{fL}{\alpha^2} \qquad C_3 = \dfrac{fL^3}{6EN_1} \\[2mm] C_4 = \dfrac{N_2\beta f(\alpha^2 L^2 \cosh \alpha L - 2\alpha L \sinh \alpha L - 2)}{2N_1 \alpha^4 \cosh \alpha L} - \dfrac{fL^4}{24EN_1} \end{cases} \tag{67}$$

## 7. Case Studies

Several examples of the simply supported single- and double-cell box beam were used to analyze the shear lag phenomenon. In order to verify the accuracy of the proposed method, the results obtained from the ANSYS finite element model and the proposed analytical theory were compared and analyzed. Finally, some recommendations about selecting displacement functions were provided.

### 7.1. Parameter Selection and Finite Element Model

The span of the simply supported beam is 40 m. The concentrated load *F* and the uniformly distributed load *f* are 200 kN and 5 kN/m, respectively. The Young's modulus *E* and the Poisson's ratio *μ* of the material are 34.5 GPa and 0.2, respectively. The shear modulus *G* can be derived from the following:

$$G = \frac{E}{2(1+\mu)} \tag{68}$$

Finite element models were constructed using the general-purpose software ANSYS 2022R2. An eight-node solid element, SOLID65, was adopted to simulate the material properties. The mesh size is 0.2 m. The loads were applied parallel to the webs uniformly to avoid torsion, distortion, and transverse bending of the cross section [15].

Figures 10a and 11a display a single-cell box girder's ANSYS model. The schematic of the cross-sectional dimensions of this single-cell box girder model is shown in Figure 1a, where $b_1$ = 3 m; $b_2$ = 2.5 m; $b_3$ = 3 m; $t_1$ = 0.2 m; $t_2$ = 0.25 m; $t_{w2}$ = 0.3 m; and *h* = 3 m.

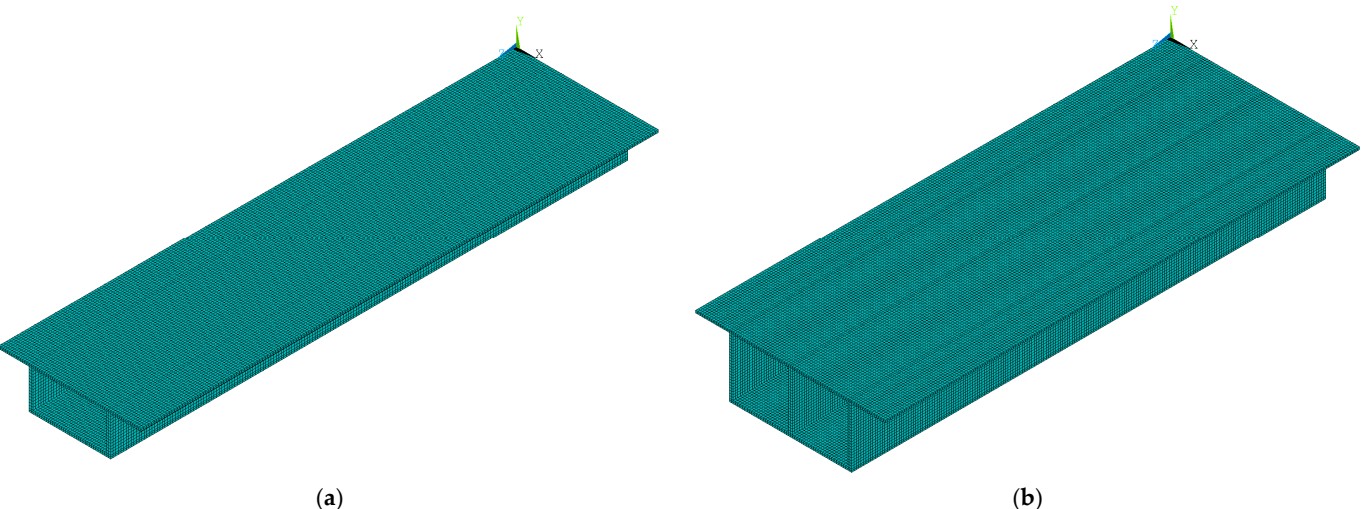

(**a**)          (**b**)

**Figure 10.** ANSYS finite element model of box girders: (**a**) single cell; (**b**) double cell.

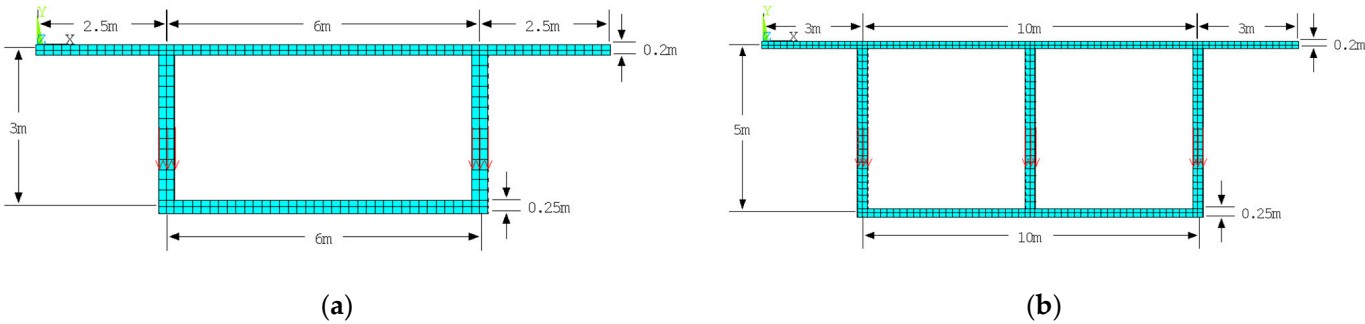

(**a**)          (**b**)

**Figure 11.** Schematic of force application in the finite element model: (**a**) single cell; (**b**) double cell.

Figures 10b and 11b display a double-cell box girder's ANSYS model. The schematic of the cross-sectional dimensions of this double-cell box girder model is shown in Figure 1b,

where $b_1 = 5$ m; $b_2 = 3$ m; $b_3 = 5$ m; $t_1 = 0.2$ m; $t_2 = 0.25$ m; $t_{w1} = 0.3$ m; $t_{w2} = 0.3$ m; and $h = 5$ m. The positions of zero points of the shear flow can be obtained according to Equation (6): $b_{11} = 2.8744$ m; $b_{12} = 2.1256$ m; $b_{21} = 3$ m; $b_{31} = 1.8115$ m; and $b_{32} = 3.1885$ m.

### 7.2. Example 1: Simply Supported Single-Cell Box Beam under Concentrated Load

Figures 12 and 13 show the vertical displacement along the girder and longitudinal distributions of shear lag coefficients for a simply supported single-cell box beam under mid-span concentrated load. It can be seen that there is a significant increase in the deflection of the beam due to the shear lag effect, and the results of displacements obtained using the proposed methods are very close to those obtained by Zhang's method [17]. In addition, the shear lag effect is clearly visible in the region surrounding the point of the concentrated force, whereas at the supports on the two sides the stresses are almost the same as those calculated using elementary beam theory. Although the results of shear lag coefficients obtained using the proposed methods and Zhang's method [17] generally agree with each other on most points, there are some differences at the mid span of the beam.

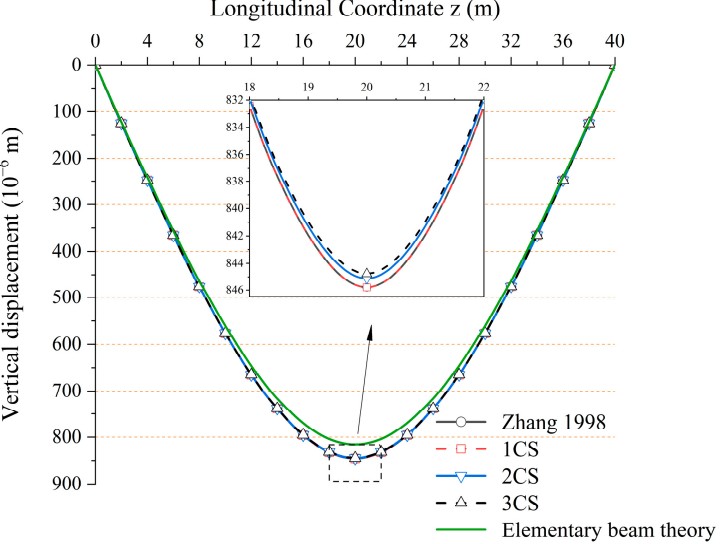

**Figure 12.** Vertical displacements of Example 1. (Zhang [17]).

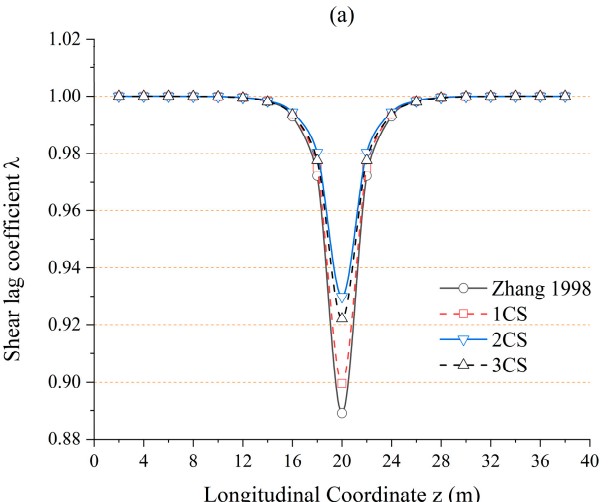

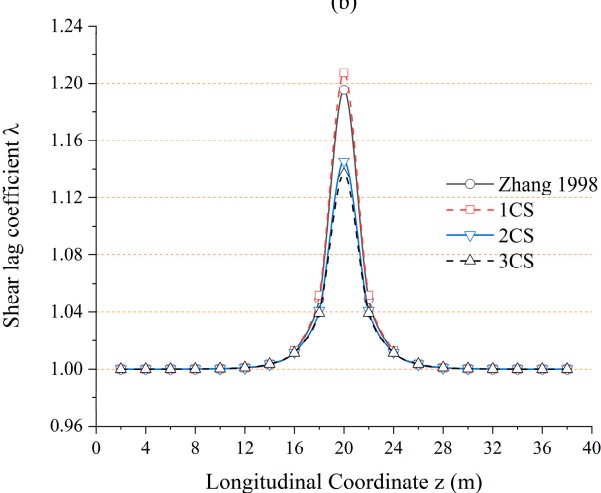

**Figure 13.** *Cont.*

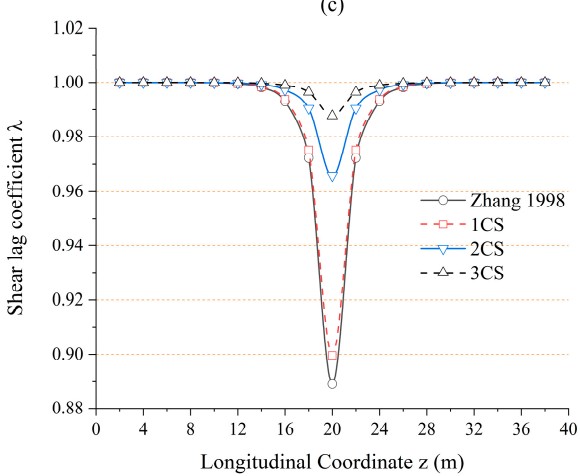

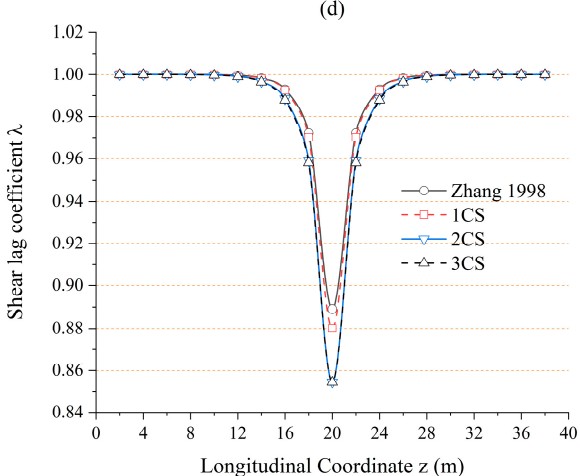

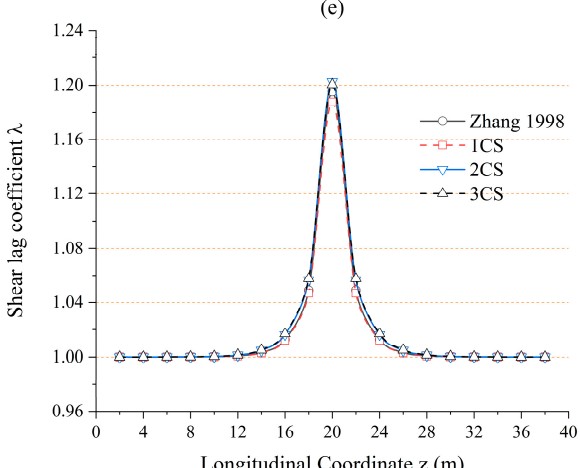

**Figure 13.** Longitudinal distributions of shear lag coefficients of Example 1: (**a**) $\lambda(a_{11})$ (top plate at $x = 0$ m); (**b**) $\lambda(a_{12})$ (top plate at $x = 3$ m); (**c**) $\lambda(a_{13})$ (cantilever plate at $x = 5$ m); (**d**) $\lambda(a_{14})$ (bottom plate at $x = 0$ m); (**e**) $\lambda(a_{15})$ (bottom plate at $x = 3$ m). (Zhang [17]).

Figure 14 shows the transverse distributions of shear lag coefficients at the mid span (20 m) and the nearby section (18 m). Table 3 lists the relative errors between the shear lag coefficient obtained by analytical methods and the results obtained by ANSYS. It can be found that the theoretical solutions fit well with the finite element solution except for the discrepancy at the point of application of the concentrated force. This problem can be attributed to the stress concentration in ANSYS. Furthermore, the results of 1*CS*, considering the axial force balance, are more accurate than those of Zhang's method [17]. The results of 2*CS* are closer to solutions of ANSYS at 18 m of the girder with a maximum relative error of only 0.06%.

**Table 3.** Relative errors of shear lag coefficients of Example 1.

| Point | ANSYS | 1*CS* | | 2*CS* | | 3*CS* | | Zhang [17] | |
|---|---|---|---|---|---|---|---|---|---|
| | | 20 m | 18 m | 20 m | 18 m | 20 m | 18 m | 20 m | 18 m |
| $a_{11}$ | | 1.04% | 0.49% | 2.29% | 0.06% | 1.44% | 0.22% | 2.20% | 0.77% |
| $a_{12}$ | | 2.02% | 0.37% | 3.21% | 1.40% | 3.96% | 1.54% | 1.01% | 0.62% |
| $a_{13}$ | 0% | 1.16% | 0.17% | 8.61% | 1.41% | 11.07% | 2.03% | 0.02% | 0.45% |
| $a_{14}$ | | 0.30% | 1.722% | 2.61% | 0.57% | 2.62% | 0.46% | 1.33% | 1.93% |
| $a_{15}$ | | 2.05% | 1.26% | 3.32% | 2.23% | 3.15% | 2.31% | 2.71% | 1.47% |

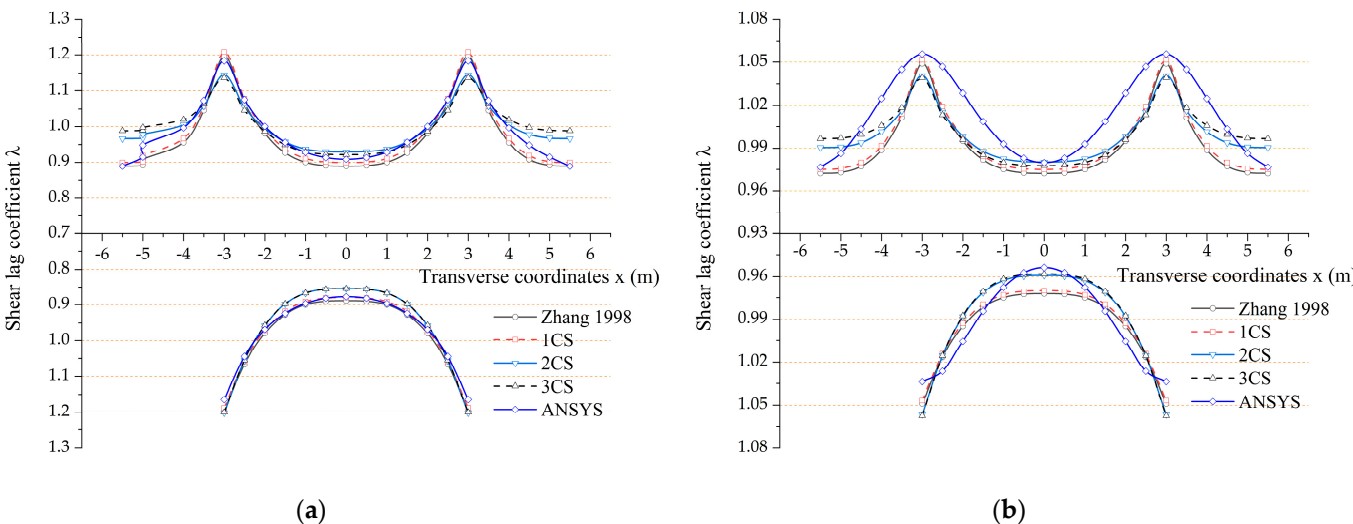

**Figure 14.** Transverse distributions of shear lag coefficients of Example 1: (**a**) *z* = 20 m; (**b**) *z* = 18 m. (Zhang [17]).

### 7.3. Example 2: Simply Supported Single-Cell Box Girder under Uniformly Distributed Load

From Figure 15, it can be seen that the results of the vertical displacements of Example 2, obtained using the proposed method, are in good agreement with those achieved using Zhang's method [17]; the maximum deflection rises by about 3.1%. Compared to Example 1, the maximum deflection of Example 2 decreases by approximately 40%, which indicates that spreading the concentrated force can effectively reduce structural deflection.

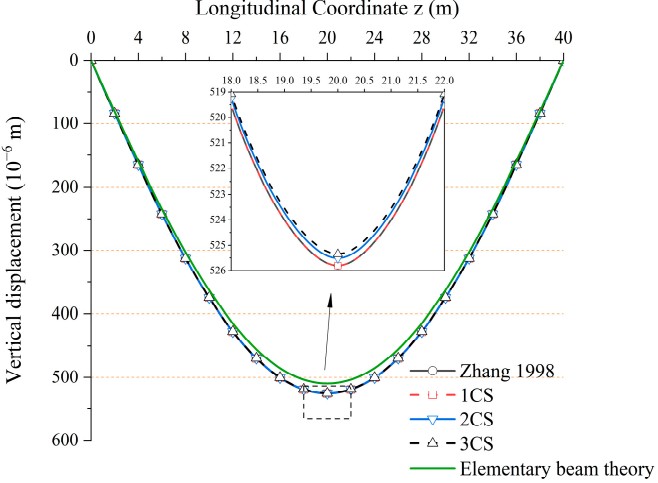

**Figure 15.** Vertical displacements of Example 2. (Zhang [17]).

As shown in Figure 16, the shear lag effect can be observed in every section of the simply supported single-cell box girder under the uniformly distributed load. Though the shear lag effect near the support is noteworthy, it can be disregarded as the minimal bending moment rarely leads to damage in practical engineering. The mid-span section (20 m) and the nearby section (18 m) with higher actual stress are selected for analysis. As seen in Figure 17 and Table 4, the results obtained by the method in this paper fit well with the finite element simulation values. It can be seen that the accuracy of Zhang's method [17] was improved.

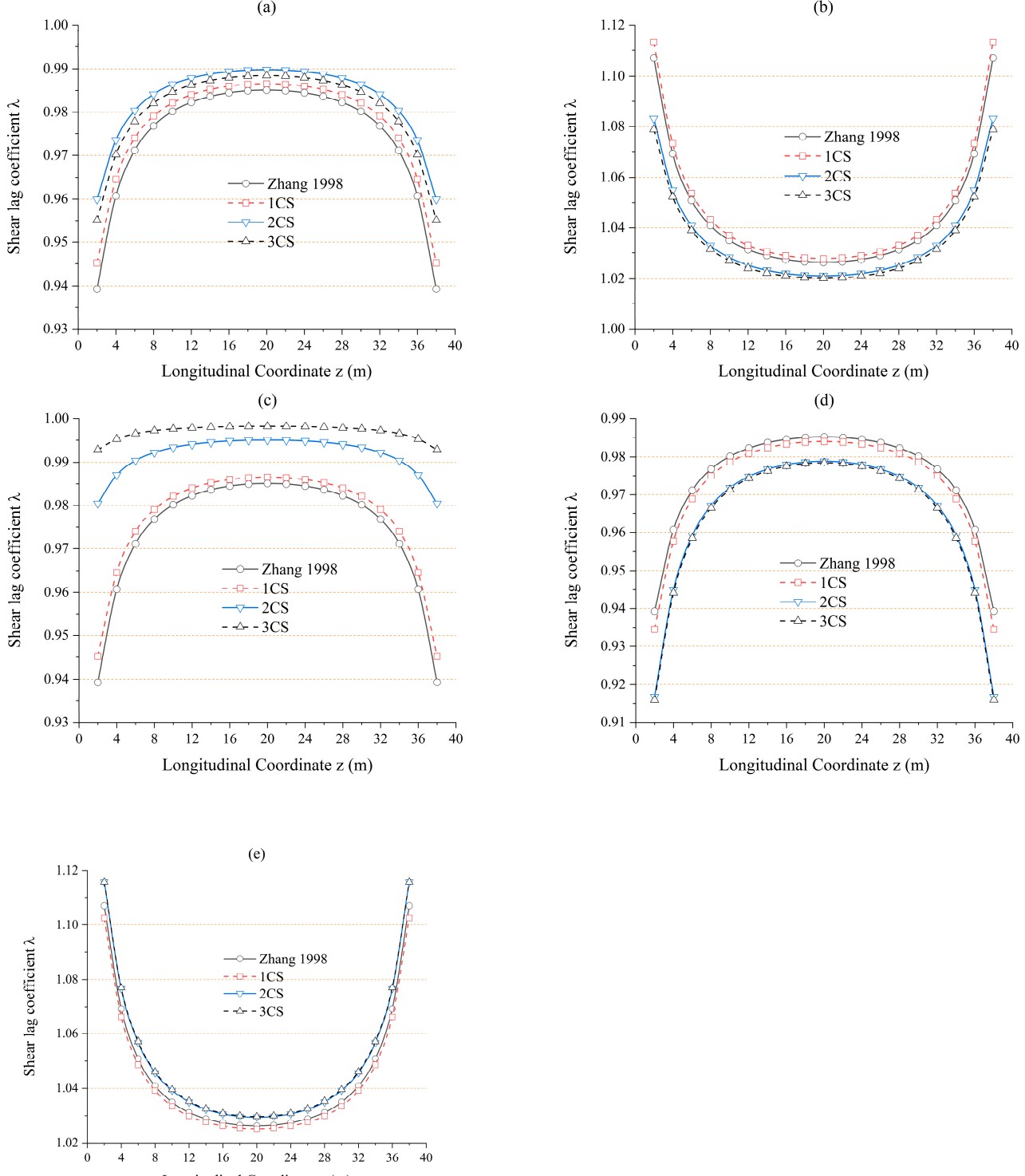

**Figure 16.** Longitudinal distributions of shear lag coefficients of Example 2: (**a**) $\lambda(a_{11})$ (top plate at $x = 0$ m); (**b**) $\lambda(a_{12})$ (top plate at $x = 3$ m); (**c**) $\lambda(a_{13})$ (cantilever plate at $x = 5$ m); (**d**) $\lambda(a_{14})$ (bottom plate at $x = 0$ m); (**e**) $\lambda(a_{15})$ (bottom plate at $x = 3$ m). (Zhang [17]).

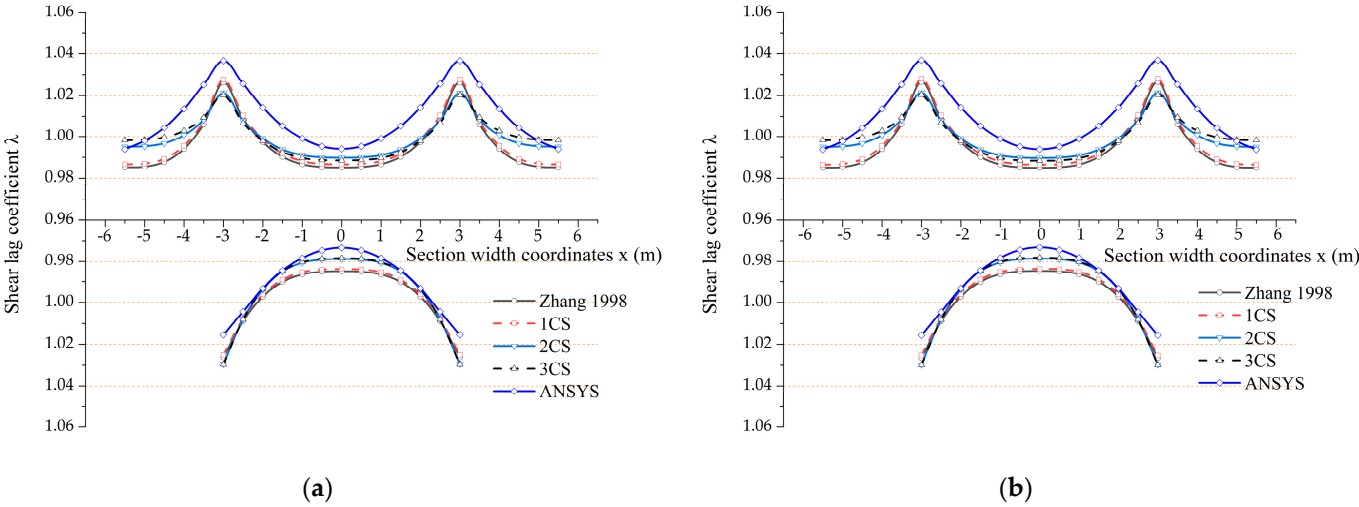

**Figure 17.** Transverse distributions of shear lag coefficients of Example 2: (**a**) *z* = 20 m; (**b**) *z* = 18 m. (Zhang [17]).

**Table 4.** Relative errors of shear lag coefficients of Example 2.

| Point | ANSYS | 1CS | | 2CS | | 3CS | | Zhang [17] | |
|---|---|---|---|---|---|---|---|---|---|
| | | 20 m | 18 m | 20 m | 18 m | 20 m | 18 m | 20 m | 18 m |
| $a_{11}$ | | 0.75% | 0.74% | 0.42% | 0.41% | 0.56% | 0.55% | 0.90% | 0.89% |
| $a_{12}$ | | 0.86% | 0.85% | 1.49% | 1.50% | 1.58% | 1.59% | 1.00% | 1.00% |
| $a_{13}$ | 0% | 0.73% | 0.73% | 0.12% | 0.13% | 0.43% | 0.45% | 0.88% | 0.88% |
| $a_{14}$ | | 1.08% | 1.08% | 0.56% | 0.55% | 0.51% | 0.51% | 1.20% | 1.20% |
| $a_{15}$ | | 0.95% | 0.95% | 1.38% | 1.38% | 1.40% | 1.40% | 1.06% | 1.06% |

From Examples 1 and 2, the 2*CS* method closely approximates the finite element solution at most points. Thus, a reasonable inference can be made that the 2*CS* method is generally more suitable for analyzing the shear lag effect of single-cell box girders.

### 7.4. Example 3: Simply Supported Double-Cell Box Beam under Concentrated Load

Figure 18 shows the vertical displacements of a simply supported double-cell box beam under the mid-span concentrated load. The shear lag coefficients obtained using the proposed method are displayed in Figures 19 and 20. Table 5 lists the relative errors of the shear lag coefficients to finite element results. The results suggest that the present method has relatively high accuracy and excellent applicability, especially the 3*CD* method. Additionally, Figures 19 and 20 demonstrate that the areas of minimum stress on the top and bottom slabs of double-cell box girders are not located at the symmetrical axis but near the zero points of shear flow. As a result, adopting the zero point of shear flow as the origin of the parabola is feasible. Furthermore, the shear lag coefficients vary at different webs. The further away from the zero point of the shear flow, the larger the value.

**Table 5.** Relative errors of shear lag coefficients of Example 3.

| Points | ANSYS | 1CD | | 2CD | | 3CD | |
|---|---|---|---|---|---|---|---|
| | | 20 m | 18 m | 20 m | 18 m | 20 m | 18 m |
| $a_{21}$ | | 0.40% | 4.64% | 3.27% | 3.32% | 6.21% | 1.26% |
| $a_{22}$ | | 4.38% | 3.51% | 0.53% | 2.86% | 1.44% | 2.59% |
| $a_{23}$ | | 2.20% | 2.37% | 0.43% | 2.62% | 2.15% | 2.54% |
| $a_{24}$ | 0% | 9.92% | 7.56% | 5.48% | 6.09% | 2.55% | 3.51% |
| $a_{25}$ | | 9.70% | 1.80% | 6.53% | 1.41% | 1.70% | 0.40% |
| $a_{26}$ | | 1.99% | 0.60% | 2.68% | 1.16% | 0.31% | 1.11% |
| $a_{27}$ | | 2.40% | 4.59% | 0.55% | 3.59% | 0.88% | 2.81% |

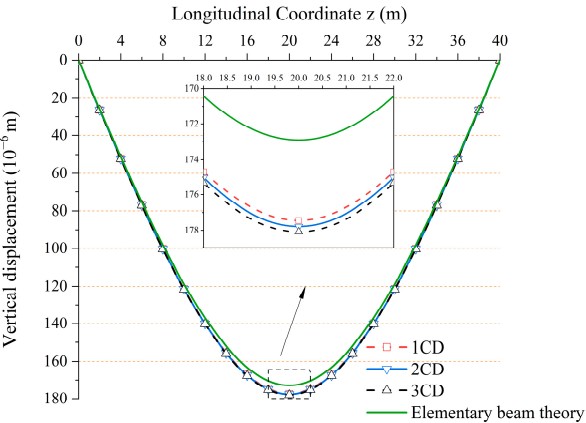

**Figure 18.** Vertical displacements of Example 3.

**Figure 19.** *Cont*.

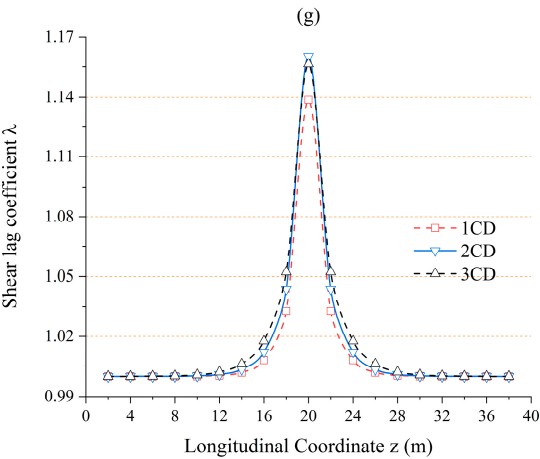

**Figure 19.** Longitudinal distributions of shear lag coefficients of Example 3: (**a**) $\lambda(a_{21})$ (top plate at $x = 0$ m); (**b**) $\lambda(a_{22})$ (top plate at $x = b_{11}$ m); (**c**) $\lambda(a_{23})$ (top plate at $x = 5$ m); (**d**) $\lambda(a_{24})$ (cantilever plate at $x = 8$ m); (**e**) $\lambda(a_{25})$ (bottom plate at $x = 0$ m); (**f**) $\lambda(a_{26})$ (bottom plate at $x = b_{31}$ m); (**g**) $\lambda(a_{27})$ (bottom plate at $x = 5$ m).

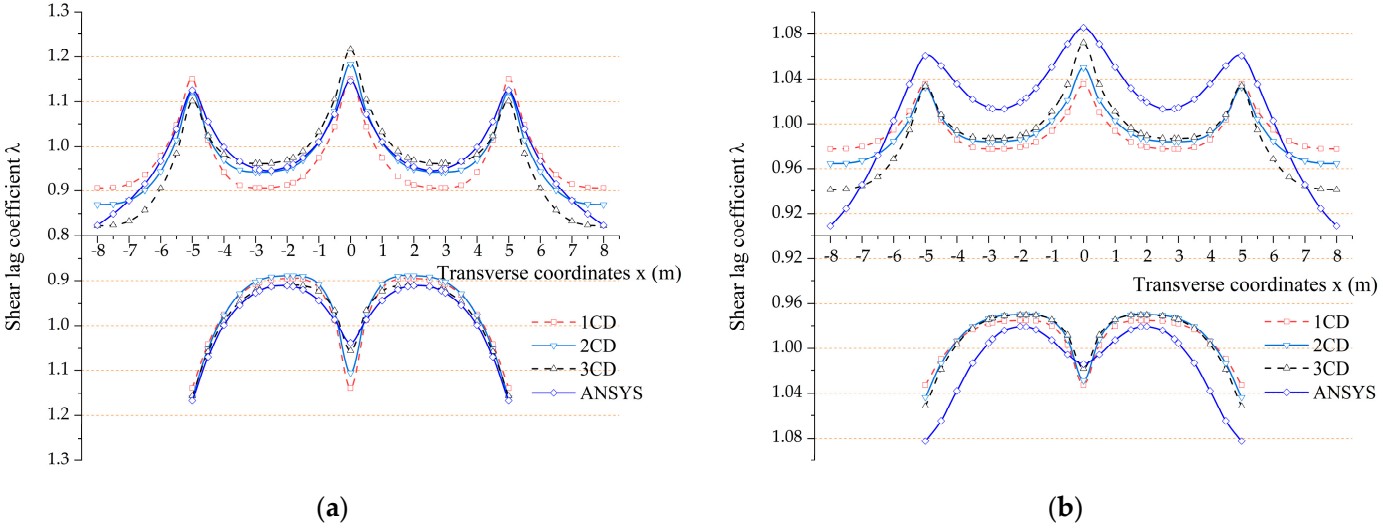

**Figure 20.** Transverse distributions of shear lag coefficients of Example 3: (**a**) $z = 20$ m; (**b**) $z = 18$ m.

### 7.5. Example 4: Simply Supported Double-Cell Box Girder under Uniformly Distributed Load

Figures 21–23 show that the shear lag effect impacts displacements and stresses of structures, and the coefficient $\eta_i$ mainly influenced the stress distribution. It can be found that the stress at the connection between the mid web and the top plate is much greater than at the junction between the side web and the top plate, but that this is reversed in the bottom plate. This is because the mid web is farther from the shear flow zero point of the top plate but closer to the shear flow zero point of the bottom plate. Therefore, the shear lag coefficient is different for different webs and is related to the distance from the web to the shear flow's zero point. Table 6 indicates that the 3*CD* method results align best with ANSYS values.

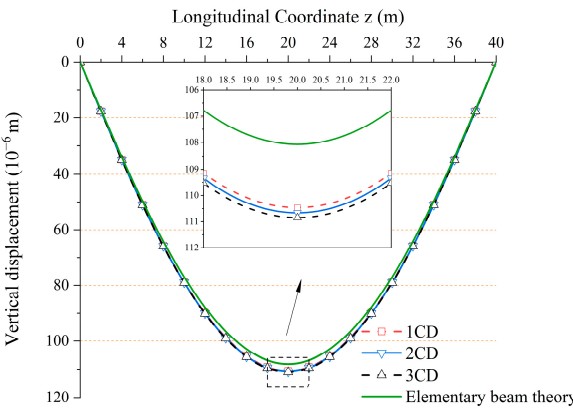

**Figure 21.** Vertical displacements of Example 4.

(a)

(b)

(c)

(d)

(e)

(f)

**Figure 22.** *Cont.*

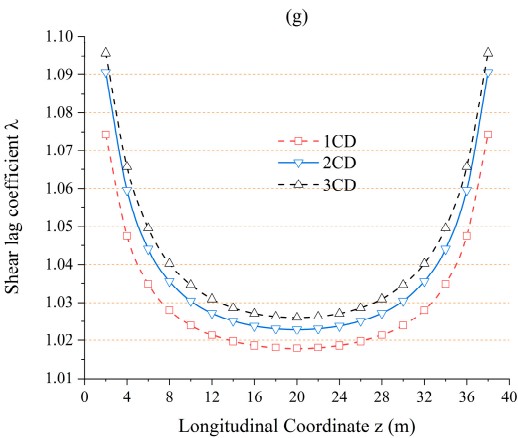

**Figure 22.** Longitudinal distributions of shear lag coefficients of Example 4: (**a**) $\lambda(a_{21})$ (top plate at $x = 0$ m); (**b**) $\lambda(a_{22})$ (top plate at $x = b_{11}$ m); (**c**) $\lambda(a_{23})$ (top plate at $x = 5$ m); (**d**) $\lambda(a_{24})$ (cantilever plate at $x = 8$ m); (**e**) $\lambda(a_{25})$ (bottom plate at $x = 0$ m); (**f**) $\lambda(a_{26})$ (bottom plate at $x = b_{31}$ m); (**g**) $\lambda(a_{27})$ (bottom plate at $x = 5$ m).

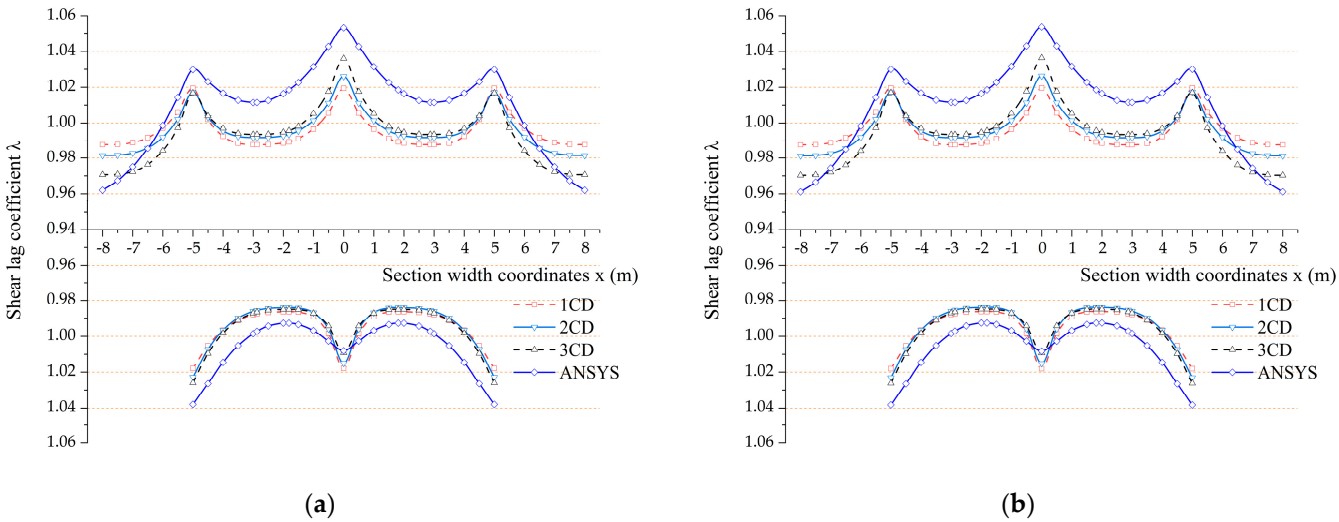

**Figure 23.** Transverse distributions of shear lag coefficients of Example 4: (**a**) $z = 20$ m; (**b**) $z = 18$ m.

**Table 6.** Relative errors of shear lag coefficients of Example 4.

| Points | ANSYS | 1CD | | 2CD | | 3CD | |
|--------|-------|------|------|------|------|------|------|
| | | **20 m** | **18 m** | **20 m** | **18 m** | **20 m** | **18 m** |
| $a_{21}$ | | 3.20% | 3.22% | 2.57% | 2.59% | 1.63% | 1.63% |
| $a_{22}$ | | 2.34% | 2.35% | 1.94% | 1.96% | 1.77% | 1.78% |
| $a_{23}$ | | 1.02% | 1.02% | 1.23% | 1.24% | 1.27% | 1.27% |
| $a_{24}$ | 0% | 2.68% | 2.77% | 2.01% | 2.09% | 0.88% | 0.95% |
| $a_{25}$ | | 0.98% | 0.98% | 0.69% | 0.69% | 0.12% | 0.11% |
| $a_{26}$ | | 0.63% | 0.63% | 0.86% | 0.87% | 0.76% | 0.77% |
| $a_{27}$ | | 1.92% | 1.93% | 1.45% | 1.46% | 1.15% | 1.15% |

From Examples 3 and 4, the 3*CD* method closely approximates the finite element solution at most points. Thus, a reasonable inference can be made that the 3*CD* method is generally more suitable for analyzing the shear lag effect of double-cell box girders.

### 7.6. Example 5: Cantilever Double-Cell Box Girder Uniformly Distributed Load

The distributions of shear lag coefficients of the cantilever double-cell box girder under uniformly distributed load are depicted in Figures 24 and 25. In this case, the negative shear

lag effect appears. Table 7 reveals that the 3*CD* method exhibits a maximum relative error of 4.69% when compared to the ANSYS values, outperforming the 1*CD* method, which yields a maximum relative error of 8.21%, and the 2*CD* method, with a maximum relative error of 6.93%.

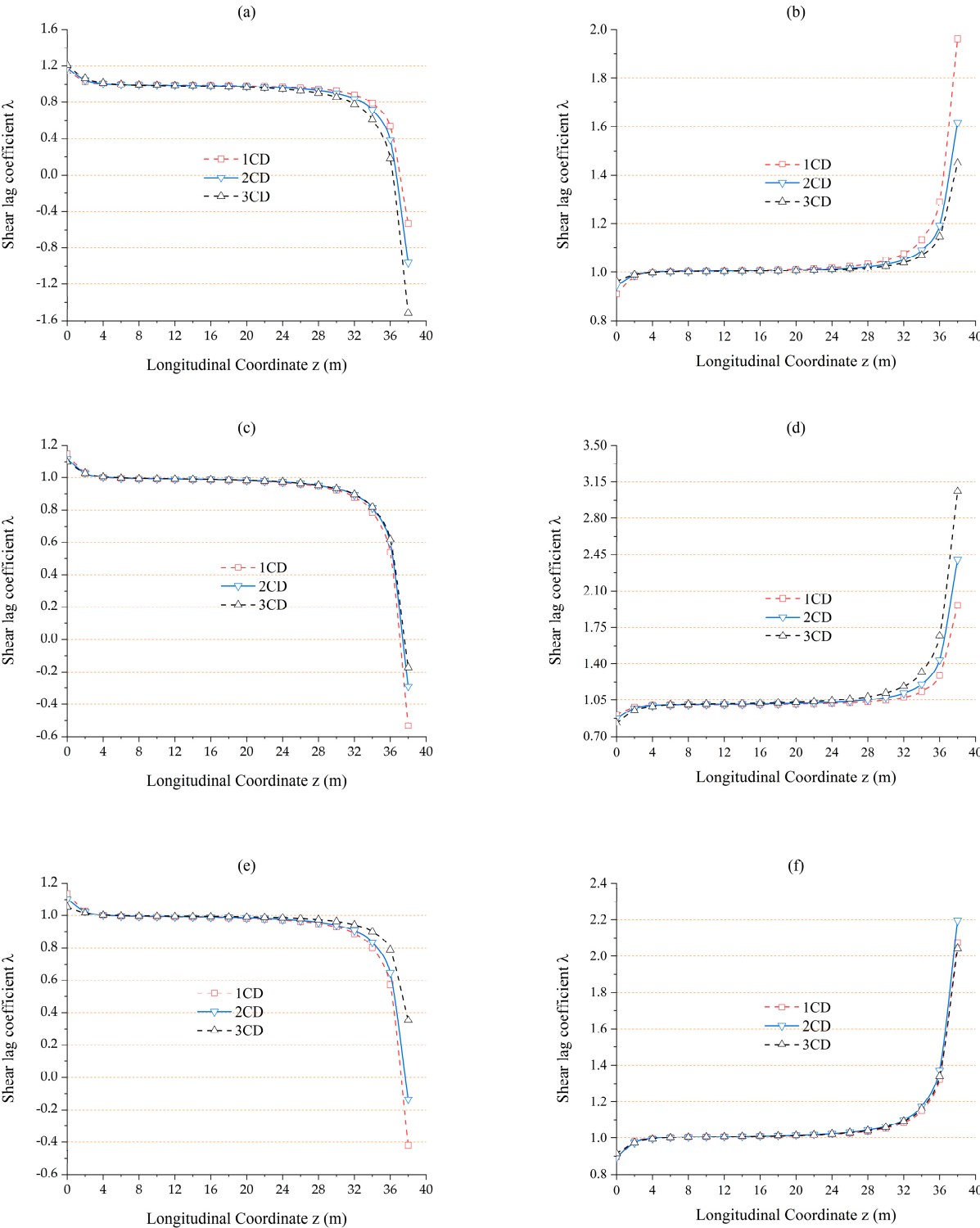

**Figure 24.** *Cont.*

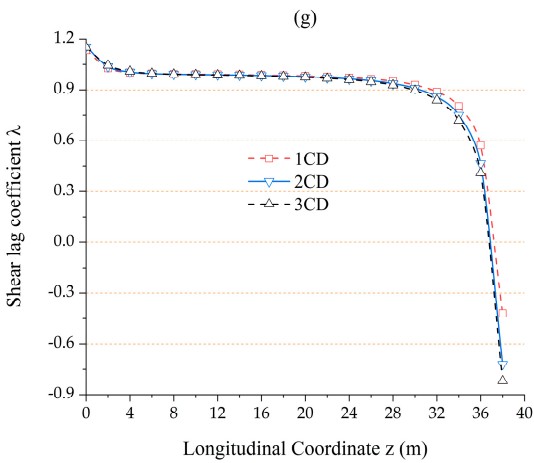

**Figure 24.** Longitudinal distributions of shear lag coefficients of Example 5: (**a**) $\lambda(a_{21})$ (top plate at $x = 0$ m); (**b**) $\lambda(a_{22})$ (top plate at $x = b_{11}$ m); (**c**) $\lambda(a_{23})$ (top plate at $x = 5$ m); (**d**) $\lambda(a_{24})$ (cantilever plate at $x = 8$ m); (**e**) $\lambda(a_{25})$ (bottom plate at $x = 0$ m); (**f**) $\lambda(a_{26})$ (bottom plate at $x = b_{31}$ m); (**g**) $\lambda(a_{27})$ (bottom plate at $x = 5$ m).

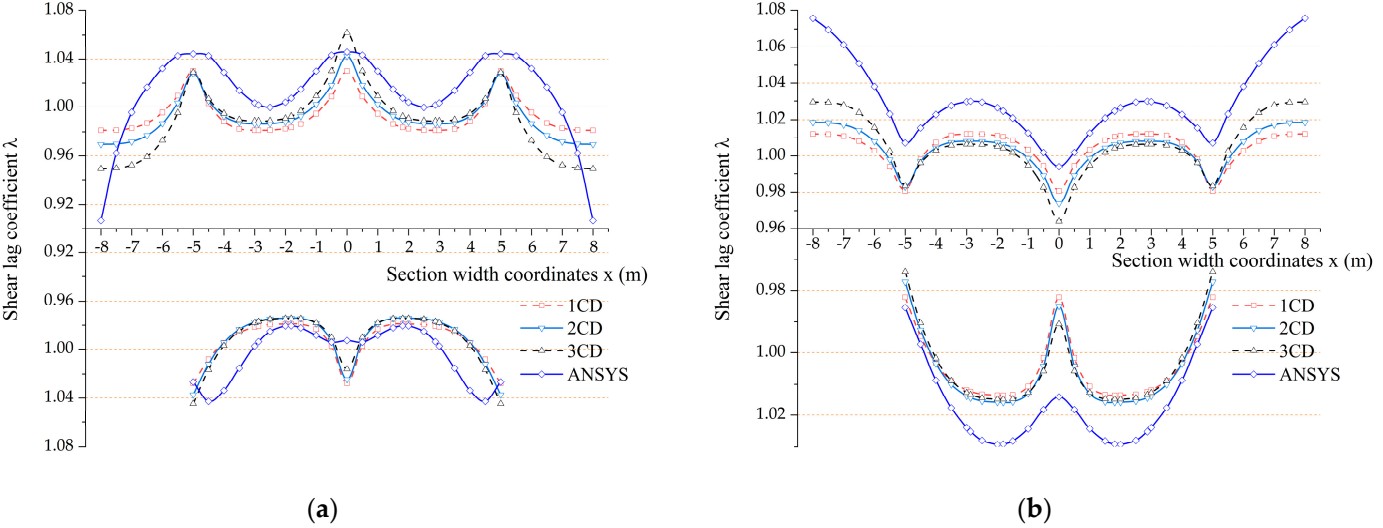

**Figure 25.** Transverse distributions of shear lag coefficients of Example 5: (**a**) $z = 2$ m; (**b**) $z = 20$ m.

**Table 7.** Relative errors of shear lag coefficients of Example 5.

| Points | ANSYS | 1CD | | 2CD | | 3CD | |
|--------|-------|--------|--------|--------|--------|--------|--------|
| | | 2 m | 20 m | 2 m | 20 m | 2 m | 20 m |
| $a_{21}$ | | 1.52% | 1.34% | 0.32% | 2.01% | 1.55% | 3.01% |
| $a_{22}$ | | 2.04% | 1.72% | 1.50% | 2.10% | 1.27% | 2.27% |
| $a_{23}$ | | 1.37% | 2.64% | 1.56% | 2.41% | 1.47% | 2.38% |
| $a_{24}$ | 0% | 8.21% | 5.91% | 6.93% | 5.31% | 4.69% | 4.30% |
| $a_{25}$ | | 3.52% | 3.16% | 3.20% | 2.87% | 2.33% | 2.30% |
| $a_{26}$ | | 0.16% | 1.51% | 0.65% | 1.29% | 0.63% | 1.39% |
| $a_{27}$ | | 0.11% | 0.34% | 1.04% | 0.83% | 1.77% | 1.15% |

### 7.7. Example 6: Double-Cell Box Girder Fixed at Both Ends under Uniformly Distributed Load

As shown in Figure 26, the beam is fixed at both ends and subjected to the uniformly distributed load. Figure 27 shows that the shear lag coefficient near the inflection point suddenly changes due to the bending moment being close to zero. Moreover, the result of the 3*CD* method is closer to the simulated value from Figure 28 and Table 8.

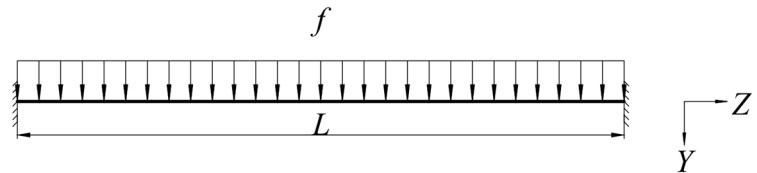

**Figure 26.** Double-cell box girder fixed at both ends under uniformly distributed load.

**Figure 27.** *Cont.*

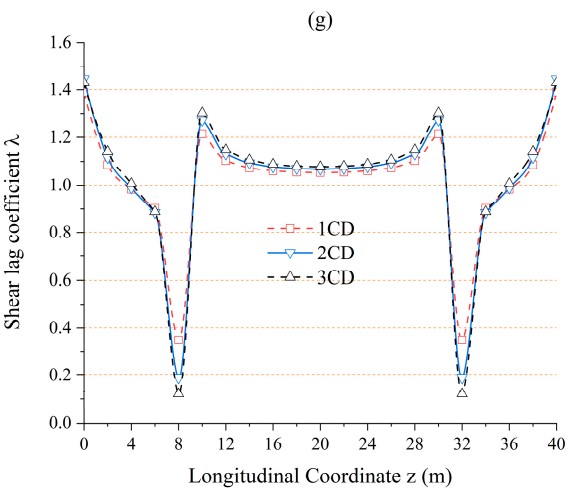

**Figure 27.** Longitudinal distributions of shear lag coefficients of Example 6: (**a**) $\lambda(a_{21})$ (top plate at $x = 0$ m); (**b**) $\lambda(a_{22})$ (top plate at $x = b_{11}$ m); (**c**) $\lambda(a_{23})$ (top plate at $x = 5$ m); (**d**) $\lambda(a_{24})$ (cantilever plate at $x = 8$ m); (**e**) $\lambda(a_{25})$ (bottom plate at $x = 0$ m); (**f**) $\lambda(a_{26})$ (bottom plate at $x = b_{31}$ m); (**g**) $\lambda(a_{27})$ (bottom plate at $x = 5$ m).

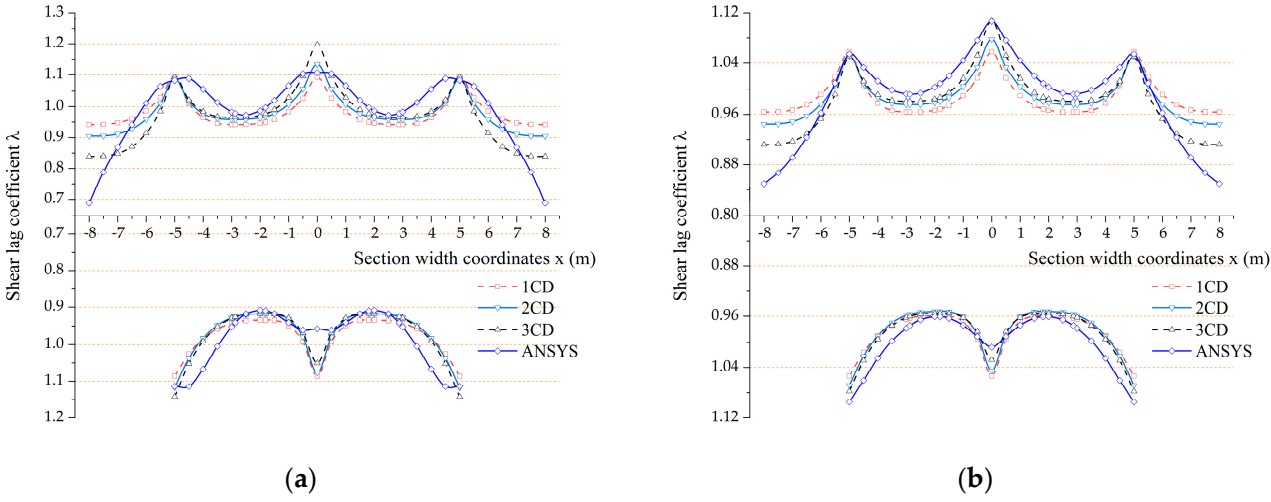

**(a)**                                          **(b)**

**Figure 28.** Transverse distributions of shear lag coefficients of Example 6: (**a**) $z = 2$ m; (**b**) $z = 20$ m.

**Table 8.** Relative errors of shear lag coefficients of Example 6.

| Points | ANSYS | 1CD | | 2CD | | 3CD | |
|---|---|---|---|---|---|---|---|
| | | **2 m** | **20 m** | **2 m** | **20 m** | **2 m** | **20 m** |
| $a_{21}$ | | 1.22% | 4.38% | 2.43% | 2.57% | 8.17% | 0.12% |
| $a_{22}$ | | 3.70% | 2.86% | 2.02% | 1.67% | 1.33% | 1.14% |
| $a_{23}$ | | 0.93% | 0.35% | 0.44% | 0.30% | 0.81% | 0.40% |
| $a_{24}$ | 0% | 36.72% | 13.41% | 31.35% | 11.14% | 21.79% | 7.31% |
| $a_{25}$ | | 13.51% | 4.59% | 12.57% | 3.73% | 9.79% | 2.00% |
| $a_{26}$ | | 2.72% | 0.13% | 0.95% | 0.85% | 0.92% | 0.55% |
| $a_{27}$ | | 2.51% | 3.76% | 0.26% | 2.42% | 2.53% | 1.55% |

## 8. Conclusions

In this study, an analytical method from the perspective of shear flow was proposed to study the shear lag effect of thin-walled single- and double-cell box girders. Longitudinal displacement functions with cubic parabola form were built using the shear flow's zero points. Then, the governing differential equations were derived through the variational energy method, and analytical solutions of the shear lag effect under several common

boundaries and load cases were presented. Finally, some cases involving the simply supported beam were used to analyze the shear lag effect. The results obtained using the proposed method were validated via comparison with numerical results. Based on the present study, the following conclusions could be drawn:

(1) The proposed method can provide reasonable predictions for the shear lag effect of single- and double-cell box girders. It is suggested that the 2$CS$ method is suited for solving the shear lag effect in single-cell box girders, and the 3$CD$ method is recommended for double-cell box girders. The analytical methods based on the shear flow distribution law proposed in this paper are more straightforward and practical. They also provide theoretical support for the subsequent development of finite beam elements.

(2) For double-cell box girders, minimum stress locations on the top and bottom slabs do not coincide with the symmetry axis. Instead, they are near the zero points of shear flow. Therefore, it is rational to adopt the zero point of shear flow as the origin of the parabola.

(3) The shear lag coefficients are varied at different webs of the double-cell box section. The magnitude of the coefficients is related to the distance from the web to the zero point of the shear flow. The further away from the zero point of the shear flow, the larger the value.

(4) Considering the web's warping variation and the section's axial force balance can increase the accuracy of the calculation results.

**Author Contributions:** Y.S., conceptualization, data curation, formal analysis, methodology, visualization, writing—original draft, and writing—review and editing; S.Z., supervision, validation, and writing—review and editing; G.W., validation; and C.Z., visualization. All authors have read and agreed to the published version of the manuscript.

**Funding:** This research received no external funding.

**Institutional Review Board Statement:** Not applicable.

**Informed Consent Statement:** Not applicable.

**Data Availability Statement:** The data presented in this study are available on request from the corresponding author. The data are not publicly available due to privacy.

**Conflicts of Interest:** The authors declare no conflicts of interest.

## Notation

$A_i$ = the area of the $i$-th flange
$b_i$ = the width of the $i$-th flange
$C_i$ = the relevant unknown coefficient determined by the boundary and continuity conditions
$d_i$ = the introduced coefficient
$E$ = the elastic modulus
$F$ = the concentrated load
$f$ = the uniformly distributed load
$f(x)$ = the distribution function corresponding to the shear lag effect
$G$ = the shear modulus
$h$ = the height of webs
$h_i$ = the distance between the centroid of the cross section and the midplane of $i$-th flange
$I_i$ = the inertial moment to the $X$-axis of the $i$-th flange
$L$ = the length of the beam
$M$ = the bending moment of the cross-section
$N_i$ = the parameter related to the cross-sectional properties
$O$ = the origin of the coordinate
$Q$ = the cross-sectional shear force
$q$ = the shear flow
$S_x$ = the static moment to the $X$-axis
$s$ = the curvilinear coordinates of the section profile
$t_i$ = the thickness of the $i$-th flange
$U_i$ = the strain energy of the $i$-th flange

$u_i(x,z)$ = the longitudinal displacement of the $i$-th flange
$V$ = the external load potential energy of the system
$w$ = the vertical deflection
$X$ = the width direction of the section
$x_0$ = the distance from the shear flow's zero point to the side web in the opened double-cell section
$Y$ = the height direction of the section
$Z$ = the longitudinal direction of the beam
$\gamma$ = the shear strain
$\varepsilon$ = the axial strain
$\eta_i$ = the introduced coefficient
$\lambda$ = the shear lag coefficient
$\mu$ = the Poisson's ratio
$\Pi$ = the total potential energy of the system
$\sigma$ = the bending normal stress
$\varphi(z)$ = the maximum difference in the shear angle

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
