# Peer review of "Analytical Method of the Shear Lag Effect in Thin-Walled Box Girders Based on the Shear Flow Distribution Law"

_applsci, doi:10.3390/app14020828_

Round 1

Reviewer 1 Report

Comments and Suggestions for Authors

The theoretical part shoul be condensed and improved.

Is there any comaprison with experiments.

Limiting cases needs to be studied.

Figures are in not good shape.

Reviewer 2 Report

Comments and Suggestions for Authors

The paper presents an analytical method to investigate the shear lag effect of thin-walled single- and double-cell box girders. The paper is interesting but some issues should be addressed: 

- State of the art should be improved, considering that few information is reported about the behaviour of thin walled box girder and the effects about this kind of girder (e.g., torsion)

- Figures are of bad quality. Please improve them

- The discussion in Section 2 seems to be not fully generalizable. I suggest to refine the description, to improve generalization

- Eq 16 makes sense if all displacements are available. The question is: are there some hypotheses to reduce available degree of freedom, i.e., displacements?

- Fig 12 and 13 are not visible, as the analogous for the other case study

- Among case studies I would like to see also a fixed beam as can occurs in some bridges where robustness is required

- Altough the advantage of a closed form solution is evident, could this approach replace the FEM solution or not? Some comments should be added to improve the merit of the work.

Reviewer 3 Report

Comments and Suggestions for Authors

1.       This paper introduces an analytical method for calculating the shear lag effect in thin-walled box girders based on the shear flow distribution law. The paper provides useful information and the results are satisfactory.

2.       The paper is generally written well. Some improvements, though, are needed to meet the publication stardard, as mentioned below.

3.       The quality of all the figures is very poor. It is recommended to redraw all the figures.

4.       The expression of Tables 1 and 2 is not clear; the magnitude is q or q*Q/I? The values in the table is S or q or q*Q/I?

5.       Fig. 6, the title does not match the figure content.

6.       Mark the dimensions in Fig. 11. What's the mesh size?

        7.       Line 520, what is "This method" referred to?

Comments on the Quality of English Language

Minor improvements are needed.

Reviewer 4 Report

Comments and Suggestions for Authors

The authors  propose an analytical method based on the shear flow distribution law to study the shear lag effect of thin-walled single- and double-cell box girders.The reliability of the method is estimated  by comparing the results obtained employing the proposed method  with numerical results supplied by ANSYS software. The paper is of average originality since the authors use a series of novel improved longitudinal displacement functions mathematically expressed  as cubic parabolas.

The few references are appropriate. Figs.12-23  are of poor quality and need to be improved.

In my opinion,the author should update the references to give an explanation of what their study findings reveal that'is unique or different from the recent analytical ones in literature. 

Comments on the Quality of English Language

The language needs minor improvement.

Round 2

Reviewer 4 Report

Comments and Suggestions for Authors

The paper has been sufficiently revised.

Comments on the Quality of English Language

Good English